# Quantitative system drift compensates for altered maternal inputs to the gap gene network of the scuttle fly *Megaselia abdita*

Karl R Wotton[1,2]*, Eva Jiménez-Guri[1,2], Anton Crombach[1,2], Hilde Janssens[1,2], Anna Alcaine-Colet[1,2,3], Steffen Lemke[4†], Urs Schmidt-Ott[4], Johannes Jaeger[1,2]*

[1]European Molecular Biology Laboratory, CRG Systems Biology Research Unit, Centre for Genomic Regulation, Barcelona, Spain; [2]Universitat Pompeu Fabra, Barcelona, Spain; [3]Universitat de Barcelona, Barcelona, Spain; [4]Department of Organismal Biology and Anatomy, University of Chicago, Chicago, United States

**Abstract** The segmentation gene network in insects can produce equivalent phenotypic outputs despite differences in upstream regulatory inputs between species. We investigate the mechanistic basis of this phenomenon through a systems-level analysis of the gap gene network in the scuttle fly *Megaselia abdita* (Phoridae). It combines quantification of gene expression at high spatio-temporal resolution with systematic knock-downs by RNA interference (RNAi). Initiation and dynamics of gap gene expression differ markedly between *M. abdita* and *Drosophila melanogaster*, while the output of the system converges to equivalent patterns at the end of the blastoderm stage. Although the qualitative structure of the gap gene network is conserved, there are differences in the strength of regulatory interactions between species. We term such network rewiring 'quantitative system drift'. It provides a mechanistic explanation for the developmental hourglass model in the dipteran lineage. Quantitative system drift is likely to be a widespread mechanism for developmental evolution.

**\*For correspondence:** karl. wotton@crg.eu (KRW); yogi. jaeger@crg.eu (JJ)

**Present address:** †Centre for Organismal Studies, Ruprecht Karls University, Heidelberg, Germany

**Competing interests:** The authors declare that no competing interests exist.

**Reviewing editor**: Naama Barkai, Weizmann Institute of Science, Israel

## Introduction

An important question for evolutionary biology is how developmental processes can compensate for variable environmental, signalling, or regulatory inputs to create a constant phenotypic outcome (*Waddington, 1942*). Segment determination during early insect embryogenesis is a well-studied example of this phenomenon. The segmentation gene network produces very robust and conserved output patterns despite fast-evolving upstream inputs through maternal gradients and vastly different modes of segmentation dynamics in different insect taxa (*Sander, 1976*; *Davis and Patel, 2002*). This type of neutral network evolution—producing the same output based on different regulatory principles—is called developmental system drift or phenogenetic drift (*Weiss and Fullerton, 2000*; *True and Haag, 2001*; *Weiss, 2005*; *Haag, 2007*; *Pavlicev and Wagner, 2012*). It is believed to be a very widespread phenomenon and can be interpreted as phenotypically neutral evolution along so-called genotype (meta-)networks. Genotype networks consist of different regulatory network structures—connected by simple mutational steps—that produce the same patterning or phenotypic output (*Ciliberti et al., 2007a*, *2007b*; *Wagner and Lynch, 2008*; *Draghi et al., 2010*; *Wagner, 2011*).

In order to discover the mechanisms underlying developmental system drift, it is necessary to systematically and quantitatively investigate the structure and dynamics of evolving regulatory networks. In this study, we use the dipteran gap gene system—constituting the first regulatory layer of the segmentation gene network (*Foe and Alberts, 1983*; *Akam, 1987*; *Ingham, 1988*; *Jaeger, 2011*)—as a model system to study developmental system drift.

**eLife digest** Similar biological phenomena can result from different processes occurring in different organisms. For example, the early stages of how an insect develops from an egg can vary substantially between different species. Nonetheless, all insects have a body plan that develops in segments. The same outcome occurring as a result of different developmental steps is known as 'system drift', but the mechanisms underlying this phenomenon are largely unknown.

How the body segments of the fruit fly *Drosophila* develop has been extensively studied. First, a female fruit fly adds messenger RNA (or mRNA) molecules copied from a number of genes into her egg cells. These mRNA molecules are then used to produce proteins whose concentration varies along the length of each egg. These proteins in turn switch on so-called 'gap genes' in differing amounts in different locations throughout the fruit fly embryo. The activity of these genes goes on to define the position and extent of specific segments along the fruit fly's body.

Like the fruit fly, the scuttle fly *Megaselia abdita* has a segmented body. However, mothers of this species deposit somewhat different protein gradients into their eggs. How the regulation of development differs in the scuttle fly to compensate for this change is unknown. Now, Wotton et al. have studied, in detail, how gap genes are regulated in this less well-understood fly species to understand the mechanisms responsible for a specific example of system drift.

In the fruit fly, gap genes normally switch-off (or reduce the expression of) other gap genes within the same developing body segment, and Wotton et al. found that the same kind of interactions tended to occur in the scuttle fly. As such, the overall structure of the gap gene network was fairly similar between scuttle and fruit flies. There were, however, differences in the strength of these interactions in the two fly species. These quantitative differences result in a different way of making the same segmental pattern in the embryo. In this way, Wotton et al. show how tinkering with the strength of specific gene interactions can provide an explanation for system drift.

The gap gene system in *Drosophila melanogaster* (family: Drosophilidae) is one of the most thoroughly studied developmental gene regulatory networks today. The genetic and molecular mechanisms of gap gene regulation have been investigated extensively over the last few decades (reviewed in *Jaeger (2011)*), and a number of mathematical models exist that faithfully reproduce gap gene expression dynamics in this species (*Reinitz et al., 1995*; *Jaeger et al., 2004a*, *2004b*; *Perkins et al., 2006*; *Ashyraliyev et al., 2009*; *Manu et al., 2009a*, *2009b*; *Crombach et al., 2012a*, *2012b*; *Becker et al., 2013*). In this study, we provide a brief summary of the most important regulatory principles that were revealed by this research.

An overview of the structure of the gap gene network is given in *Figure 1* (grey inset). Gap genes receive regulatory inputs from maternal protein gradients formed by the gene products of *bicoid* (*bcd*), *hunchback* (*hb*), and *caudal* (*cad*) (*Figure 1*, top row of graphs) (reviewed in *St Johnston and Nüsslein-Volhard (1992)*). These gradients set up an initial asymmetry along the major or antero-posterior (A–P) axis of the embryo. Bcd and Cad activate the four trunk gap genes *hb*, *Krüppel* (*Kr*), *knirps* (*kni*), and *giant* (*gt*), which become expressed in broad, overlapping domains (*Figure 1*, bottom row of graphs). As development proceeds, gap gene cross-repression refines the pattern (*Figure 1*, grey inset) (see *Jaeger, 2011* for review). Domains of *hb* and *kni* as well as *Kr* and *gt* have mutually exclusive expression patterns and show strong mutual repression. This 'alternating cushions' mechanism maintains and sharpens the basic staggered arrangement of gap domains. The expression patterns of *Kr/kni*, *kni/gt*, and *gt/hb* overlap and show weaker repression with a posterior-to-anterior bias, which results in anterior shifts of each of these domains over time. Finally, trunk gap gene expression is repressed in the posterior pole region of the embryo by the terminal gap genes *tailless* (*tll*) and *huckebein* (*hkb*).

Our understanding of the dipteran gap gene system outside the drosophilid family is much less well developed. This limitation applies in particular to nematoceran midges and mosquitoes (*Figure 1*, black phylogenetic branches on the right) (*Jiménez-Guri et al., 2013*). Qualitative (*Rohr et al., 1999*; *García-Solache et al., 2010*; *Jiménez-Guri et al., 2014*) and quantitative (*Crombach et al., 2014*; *Janssens et al., 2014*) studies of segmentation gene expression in the moth midge *Clogmia albipunctata* (family: Psychodidae) indicate that posterior patterning—especially the onset of posterior

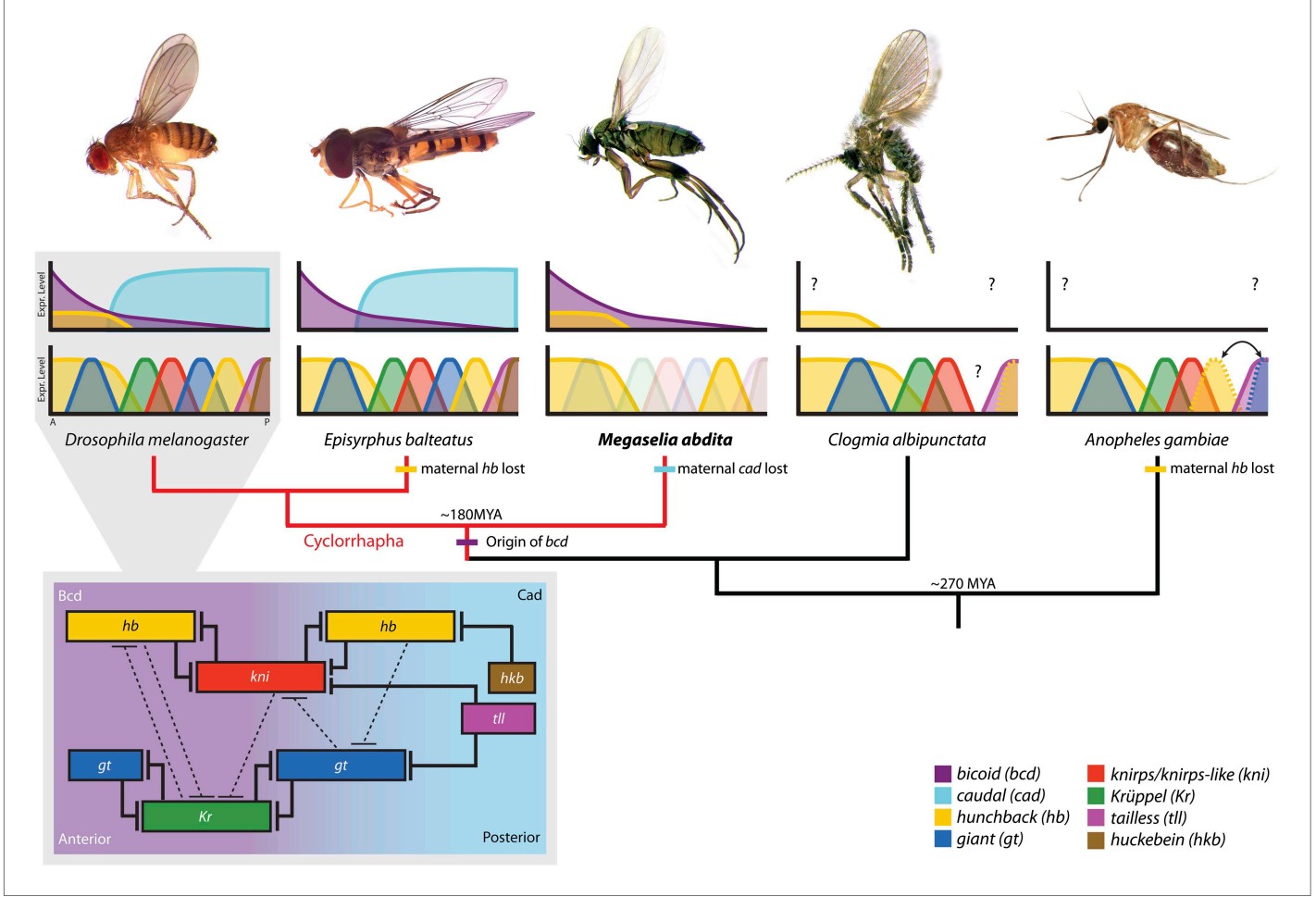

**Figure 1**. The evolution of the dipteran gap gene network. A simplified phylogenetic tree of the order Diptera indicates the relative position of *M. abdita* with regard to other species of flies, midges, and mosquitoes in which gap genes have been studied in some detail. *M. abdita* belongs to the brachyc-eran infra-order Cyclorrhapha (marked in red); paraphyletic nematoceran lineages are shown in black. Only cyclorrhaphan flies have a *bcd* gene. Other maternal gradients have been lost along various branches of the tree (as indicated). For each species, we show an image of the adult (top), as well as a schematic representation of the spatial arrangement of maternal gradients (middle), and gap gene expression domains (bottom). Y-axes show expression levels (in normalised arbitrary units); X-axes show position along the major axis of the embryo (A: anterior, P: posterior). See key for colour coding. Solid colours indicate previously published expression patterns; faded colours represent previously unknown patterns reported in this study. Question marks indicate unknown maternal gradients or potentially missing gap domains. The inset (grey background) shows the gap gene network in *D. melanogaster*. Within the inset, background colour indicates major maternal regulatory inputs, boxes show the position of gap domains along the A–P axis; T-bars represent strong (solid) or weak (dashed) cross-repressive interactions among gap genes. Species (families): *D. melanogaster* (Drosophilidae): vinegar fly; *E. balteatus* (Syrphidae): marmalade hoverfly; *M. abdita* (Phoridae): scuttle fly; *C. albipunctata* (Psychodidae): moth midge; *A. gambiae* (Culicidae): malaria mosquito. All images are our own except *A. gambiae* image taken by Muhammad Mahdi Karim (source: Wikimedia Commons). See text for details.

*hb* expression—is delayed and that repression between *Kr* and *gt* may not be conserved in this species, which lacks a posterior *gt* domain. A similar posterior delay is observed in the malaria mosquito *Anopheles gambiae* (Culicidae), where both posterior domains of *hb* and *gt* are present, but appear only after gastrulation in an A–P order which is reversed compared to that observed in *D. melanogaster* (**Goltsev et al., 2004**). Unfortunately, several factors involved in gap gene regu-lation are still unknown in these nematoceran species, and the lack of functional evidence based on genetic perturbations limits the interpretation of experimental results and verification of net-work models.

The situation is slightly better within the Cyclorrhapha, a sub-taxon of brachyceran or 'higher' flies (**Figure 1**, red phylogenetic branches on the left) (**Wiegmann et al., 2011**; **Jiménez-Guri et al., 2013**).

Apart from *D. melanogaster*, two other cyclorrhaphan species are emerging as experimentally tractable models. One of these is the marmalade hoverfly *Episyrphus balteatus* (Syrphidae). In this species, the pattern and order of gap gene activation along the A–P axis is largely conserved compared to *D. melanogaster*, even though there is no maternal contribution to *hb* expression (*Figure 1*) (*Lemke and Schmidt-Ott, 2009*; *Lemke et al., 2010*). However, the precise dynamics of gap gene expression and the nature of gap gene cross-regulation have not yet been investigated in this species.

The second emerging non-drosophilid model system is *Megaselia abdita*, a member of the early-branching cyclorrhaphan family Phoridae (*Figure 1*) (*Wiegmann et al., 2011*; *Rafiqi et al., 2011a*; *Jiménez-Guri et al., 2013*). In what follows, we will focus on this species. Gap gene expression in *M. abdita* is less well studied than in *E. balteatus* (*Figure 1*) and little is known about gap gene regulation. Maternal transcripts of *bcd* and *hb* can be detected (*Stauber et al., 1999*, *2000*; *Lemke et al., 2008*), but no maternal contribution to *cad* expression is present in this species (*Stauber et al., 2008*). Localised maternal *bcd* transcripts at the anterior pole extend further posterior, and *bcd* RNAi cuticle phenotypes affect more posterior segments than in *D. melanogaster*, indicating a broader role of Bcd in *M. abdita* (*Stauber et al., 1999*, *2000*). In contrast, expression of maternal and zygotic *hb* appears very similar in both species (*Stauber et al., 2000*; *Lemke et al., 2008*). The zygotic anterior *hb* domain is severely depleted in embryos treated with *bcd* RNAi, suggesting activation of *hb* by Bcd as in *D. melanogaster* (*Lemke et al., 2008*). Moreover, the regulatory role of *hb* seems to be similar as well, since cuticle phenotypes of *hb* RNAi-treated *M. abdita* embryos resemble *hb* hypomorphs in *D. melanogaster* (*Stauber et al., 2000*). Finally, *Kr* expression has been mentioned to be similar in both species (*Rafiqi et al., 2008*), although no data were shown since the study focused on extraembryonic tissues at later stages of development.

The similarity of gap gene expression across cyclorrhaphan species—despite considerable variation in maternal gradients—provides an excellent opportunity to study developmental system drift in an experimentally tractable set of laboratory models. A comparative, network-level comparison of gap gene regulation would reveal how the network compensates for variable upstream inputs and inter-species differences in initial expression patterns. As a foundation for such an analysis we have previously sequenced the early embryonic transcriptome of *M. abdita* (*Jiménez-Guri et al., 2013*), and have established a precise staging scheme for embryogenesis—with a particular focus on the blastoderm stage (*Wotton et al., 2014*).

In this paper, we present a comprehensive analysis of gap gene expression in *M. abdita*, based on quantitative data with high spatial and temporal resolution. We compare wild-type expression dynamics to an equivalent dataset from *D. melanogaster* (*Crombach et al., 2012a*). Our comparison reveals that gap gene expression differs significantly between the two species during early to mid blastoderm stage but converges to equivalent patterns before the onset of gastrulation. In particular, we find that the broader influence of Bcd in *M. abdita* causes gap domains to emerge at more posterior positions compared to *D. melanogaster*. In addition, the absence of maternal Cad causes a delay in the onset of anterior shifts of posterior gap domains in *M. abdita*. This delay is later compensated by stronger shifts compared to *D. melanogaster*. To investigate the regulatory changes underlying these altered dynamics, we use previously established RNAi protocols (*Stauber et al., 2000*; *Lemke et al., 2008*) to systematically knock-down trunk and terminal gap genes. We then assay expression of the remaining trunk gap genes in each genetically perturbed background using the same high-resolution quantitative approach as for wild-type patterns. Our knock-down analysis reveals that qualitative regulatory structure—the type of interactions between gap genes—is strongly conserved between species. We do, however, detect changes significantly affecting the strength of gap gene cross-repression. Based on this, we suggest quantitative changes in regulatory mechanisms for the altered dynamics of gap gene expression in *M. abdita*. Our evidence suggests that such quantitative developmental system drift is a common feature of evolving developmental systems.

## Results and discussion

### A quantitative atlas of maternal co-ordinate and gap gene expression in *M. abdita*

We took images of *M. abdita* blastoderm-stage embryos stained for maternal co-ordinate and gap gene mRNAs as previously described (*Crombach et al., 2012a*). Embryos were classified by cleavage cycle (C10–14A), and eight separate time classes within C14A (T1–8), according to the staging scheme

in *Wotton et al. (2014)*. This results in a temporal resolution for our expression data of around 20 min during C13 and 7 min for each time class during C14A. Embryo images were processed as described in *Crombach et al. (2012b)* to extract the position of expression domain boundaries (*Figure 2A*). *Table 1* and *Supplementary file 1* summarise the number of embryos in our dataset. We then determined median boundary positions and expression variability for each gene at each time point across embryos, resulting in a quantitative, integrated spatio-temporal dataset for gap gene expression in *M. abdita* (*Figure 2B*). Sample size and resolution of this dataset are comparable to our previously published gap gene mRNA expression data for *D. melanogaster* (*Table 1*, *Supplementary file 1*) (*Crombach et al., 2012a*) and are much higher than those achieved for our quantitative dataset of gap gene expression in another non-model dipteran, the moth midge *Clogmia albipunctata* (*Crombach et al., 2014*). All the expression patterns presented here are available online through the SuperFly database (http://superfly.crg.eu) (*Cicin-Sain et al., 2015*).

Figure 3 shows the expression patterns of the trunk gap genes *hb, Kr, kni,* and *gt* in *M. abdita* and compares them to the equivalent patterns in *D. melanogaster*. We provide numerical values for boundary positions, domain widths, boundary shifts, and domain overlaps between *M. abdita* and *D. melanogaster* in *Supplementary file 2A–D. Box 1* provides a brief discussion of the most salient expression features of maternal co-ordinate and gap genes in both species. More detailed descriptions and figures characterising gene expression patterns, as well as more detailed references to the *Drosophila* literature, are provided in Appendix I below. Tables with measured boundary positions can be found in *Supplementary file 3*.

In summary, gap gene expression is qualitatively similar to *M. abdita* compared to *D. melanogaster*, yet shows measurable quantitative differences (*Figure 3A,B*). On the one hand, the relative positioning and timing of expression domains with respect to each other is largely conserved. On the other hand, the dynamics of expression differ significantly between the species (*Figure 3B*). In *D. melanogaster*, the posterior boundary of the anterior *hb* domain is stable, while this border shifts to the anterior in *M. abdita*, which has a stable posterior domain of anterior *gt* instead. Moreover, in the posterior region of the embryo, gap genes are initiated more posteriorly in *M. abdita* compared to *D. melanogaster*, and patterning is delayed with posterior gap domains retracting from the pole and shifting to the anterior at later stages in *M. abdita*. This delay is compensated by very rapid anterior shifts of posterior gap domains (especially *gt*) during the second half of the blastoderm stage. Finally, gap domains (the central *Kr* domain in particular) are wider in *M. abdita* than in *D. melanogaster*. This leads to greater overlaps between adjacent expression domains.

## Altered distribution of *bcd* and *cad* leads to posterior patterning delays

The extended range of *bcd* in *M. abdita* (*Stauber et al., 1999*, *2000*) is consistent with the observed posterior displacement of initial gap domains during the early blastoderm stage, which resembles expression patterns in embryos with increased *bcd* dosage in *D. melanogaster* (*Driever and Nüsslein-Volhard, 1988*; *Driever et al., 1989*; *Struhl et al., 1989*). To test if the placement of gap domains depends on Bcd levels in *M. abdita* (as in *D. melanogaster*), we stained embryos treated with *bcd* RNAi and processed them through the same pipeline as wild-type embryos to measure the position of gene expression boundaries (*Figure 2* and *Figure 4*). Variability in the efficacy of gene knock-downs in different embryos yields a result similar to an allelic series in classical genetics, and thus allows us to explicitly assess anterior *hb* boundary sensitivity to Bcd levels. In *Figure 4E*, we plot individual gene expression boundaries from RNAi-treated embryos against the positions of the equivalent wild-type borders (see also *Figure 2C*). We find that reduced levels of *bcd* in RNAi embryos result in an increasingly anterior position of the *hb* boundary in *M. abdita* (*Figure 4*), suggesting that the placement of gap domains is dependent on Bcd levels as in *D. melanogaster*.

The observed delay in posterior gap patterning could be linked to differences in *cad* expression between the two species: in particular, the absence of maternal *cad* and/or altered levels of zygotic *cad* gene products in *M. abdita*. We assess this possibility by creating germ-line clones (GLC) in *D. melanogaster*. The resulting embryos lack maternal, but retain one copy of zygotic, *cad*. They exhibit early initiation of posterior *gt* and *kni* expression within normal wild-type variability. However, as development proceeds during cleavage cycle 14A (C14A) these domains show substantial posterior displacement compared to the wild-type (*Figure 5*). This suggests that altered levels and timing of *cad* expression are indeed responsible for the delayed shifts of posterior gap domains in *M. abdita*.

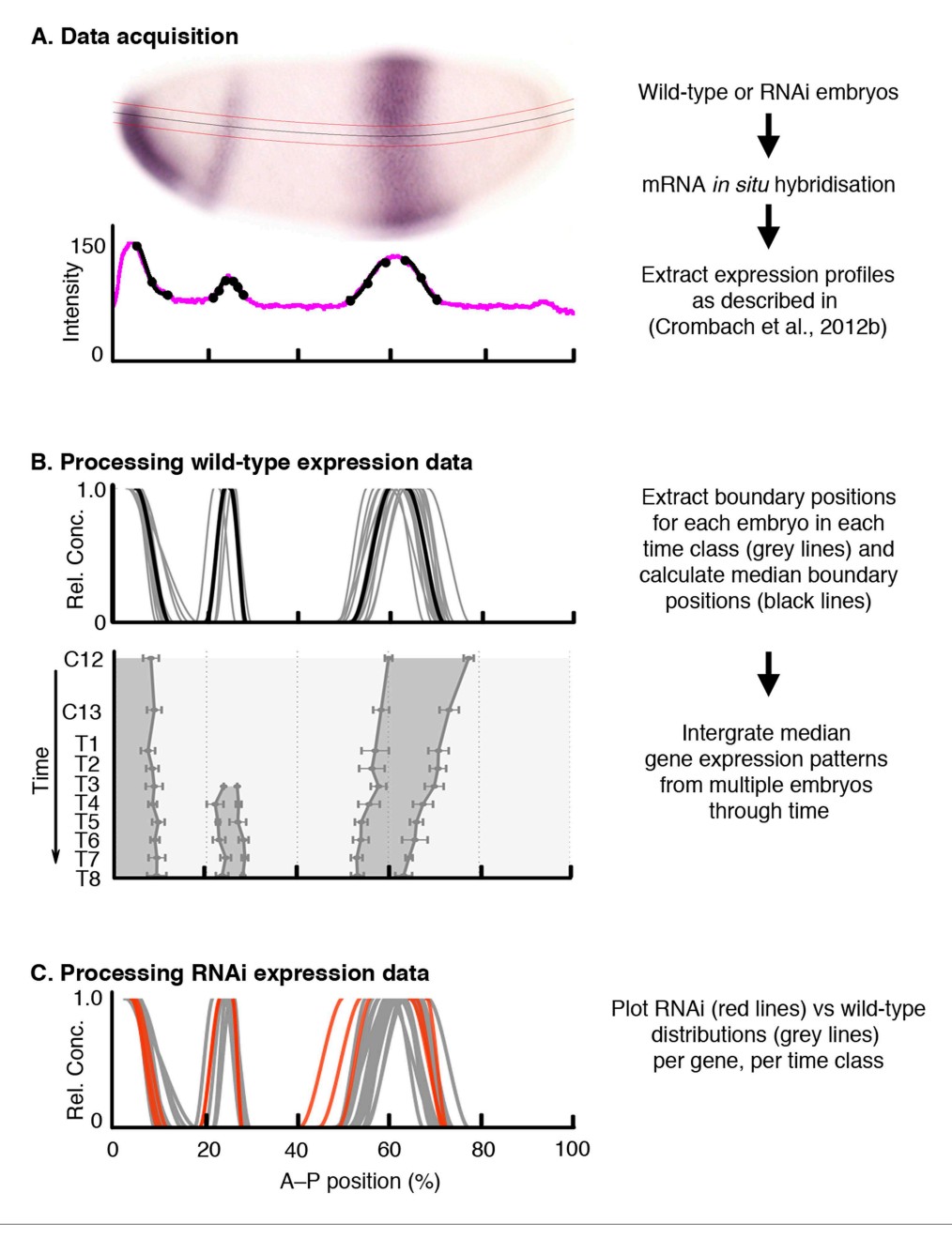

**Figure 2**. Data acquisition and processing of wild-type and RNAi knock-down embryos. As an example, we show *kni* mRNA expression in wild-type and *hb* RNAi-treated embryos. (**A**) Data acquisition: wild-type or RNAi-treated embryos were stained by single or double in situ hybridisation using an enzymatic (colorimetric) protocol as described in ***Crombach et al. (2012a)***. Embryo image shows a lateral view (anterior to the left, dorsal up) stained for *kni* mRNA (purple). We extract boundary positions as described in ***Crombach et al. (2012b)***: first, we determine a 10% strip (delimited by red lines) along the midline of the dorso-ventral axis (black line). After the extraction of the intensity profile within this strip (magenta line in graph), we manually fit clamped splines to domain boundaries (black lines with mid- and end-points indicated by circles). (**B**) Extracted boundaries from wild-type embryos are classified by gene and time class (grey lines) and used to calculate median boundary positions (black line in upper panel). This yields an integrated spatio-temporal dataset of gene expression (lower panel). Shaded area represents regions of active expression delimited by positions of half-maximal expression for median boundaries (solid lines). Error bars represent 1.5x median absolute deviation (MAD), which approximates one standard deviation. (**C**) RNAi expression data (red) are plotted against wild-type boundary positions (grey). X-axes represent % A–P position
*Figure 2. Continued on next page*

*Figure 2. Continued*

(where 0% is the anterior pole). Y-axes represent pixel intensity or relative mRNA concentration in arbitrary units, except for the lower panel in (**B**), where the Y-axis represents time flowing downwards. C12/13: cleavage cycles 12/13; T1–8: time classes within C14A as defined in *Wotton et al. (2014)*.

## Systematic investigation of gap gene cross-regulation by RNAi knock-down

To investigate the role of gap–gap cross-regulatory interactions in the patterning of the *M. abdita* embryo, we systematically knocked down each trunk and terminal gap gene by embryonic RNAi. For the trunk gap genes, the resulting severe cuticle phenotypes are similar to those observed in the equivalent null mutants of *D. melanogaster* (see Appendix II below for detailed descriptions). We stained RNAi-treated embryos for each of the trunk gap genes and processed them through the same pipeline as wild-type embryos to measure the position of gene expression boundaries (*Figure 2A,C*) (*Crombach et al., 2012b*). The resulting RNAi datasets are summarised in *Table 2* and documented in

**Table 1.** mRNA expression datasets for *M. abdita.* and *D. melanogaster*

| Domain | ant *hb* | ant *gt* | | *Kr* | | *kni* | | post *gt* | | post *hb* | |
|---|---|---|---|---|---|---|---|---|---|---|---|
| Boundary | P | A | P | A | P | A | P | A | P | A | P |
| C11 | 4 | - | - | - | - | - | - | - | - | - | - |
| | - | - | - | - | - | - | - | - | - | - | - |
| C12 | 22 | 2 | 2 | 8 | 7 | 4 | 4 | 4 | - | 2 | - |
| | 4 | - | - | 1 | 1 | - | 1 | 1 | - | - | - |
| C13 | 8 | 11 | 11 | 7 | 7 | 14 | 14 | 11 | 2 | 5 | - |
| | 31 | 8 | 8 | 9 | 6 | 16 | 14 | 11 | - | 1 | - |
| T1 | 8 | 6 | 5 | 9 | 8 | 10 | 10 | 7 | 3 | 7 | - |
| | 15 | 14 | 14 | 10 | 7 | 15 | 13 | 11 | 10 | 5 | - |
| T2 | 7 | 7 | 7 | 7 | 7 | 19 | 19 | 7 | 4 | 8 | - |
| | 13 | 8 | 10 | 6 | 5 | 19 | 16 | 8 | 9 | 8 | 1 |
| T3 | 19 | 14 | 13 | 8 | 8 | 12 | 12 | 15 | 10 | 18 | - |
| | 16 | 11 | 15 | 9 | 5 | 17 | 15 | 14 | 17 | 11 | 6 |
| T4 | 4 | 11 | 13 | 10 | 9 | 14 | 14 | 13 | 13 | 4 | - |
| | 7 | 11 | 12 | 10 | 4 | 18 | 17 | 9 | 13 | 10 | 7 |
| T5 | 5 | 8 | 8 | 8 | 7 | 11 | 11 | 9 | 8 | 4 | - |
| | 12 | 16 | 17 | 16 | 11 | 18 | 17 | 14 | 14 | 10 | 8 |
| T6 | 4 | 10 | 10 | 7 | 7 | 8 | 8 | 10 | 10 | 4 | 3 |
| | 9 | 14 | 14 | 17 | 15 | 15 | 15 | 14 | 12 | 11 | 14 |
| T7 | 2 | 7 | 7 | 13 | 13 | 8 | 8 | 6 | 6 | 2 | 2 |
| | 5 | 8 | 8 | 11 | 11 | 5 | 5 | 8 | 6 | 10 | 9 |
| T8 | 8 | 9 | 8 | 6 | 6 | 6 | 6 | 7 | 5 | 8 | 8 |
| | 8 | 12 | 12 | 6 | 6 | 6 | 6 | 7 | 6 | 9 | 8 |

This table shows the number of embryos used to calculate median positions for each expression boundary at each point in time (*M. abdita*: white rows; *D. melanogaster*: grey rows). Ant: anterior, Post: posterior domain. A indicates anterior, P posterior boundary of a domain. Time classification as defined in *Wotton et al. (2014)* for *M. abdita* and in *Surkova et al. (2008b)* for *D. melanogaster*: C11–13 correspond to cleavage cycles 11 to 13; T1–8 represent time classes subdividing C14A. Our *M. abdita* expression dataset consists of a total of 367 embryos (91 stained for *hb*, 83 for *Kr*, 87 for *gt*, and 106 for *kni*). An additional 115 embryos make up the dataset for maternal co-ordinate and terminal gap genes shown in *Supplementary file 1*. The *D. melanogaster* gap gene dataset has been published previously (*Crombach et al., 2012a*). It is included here for comparison.

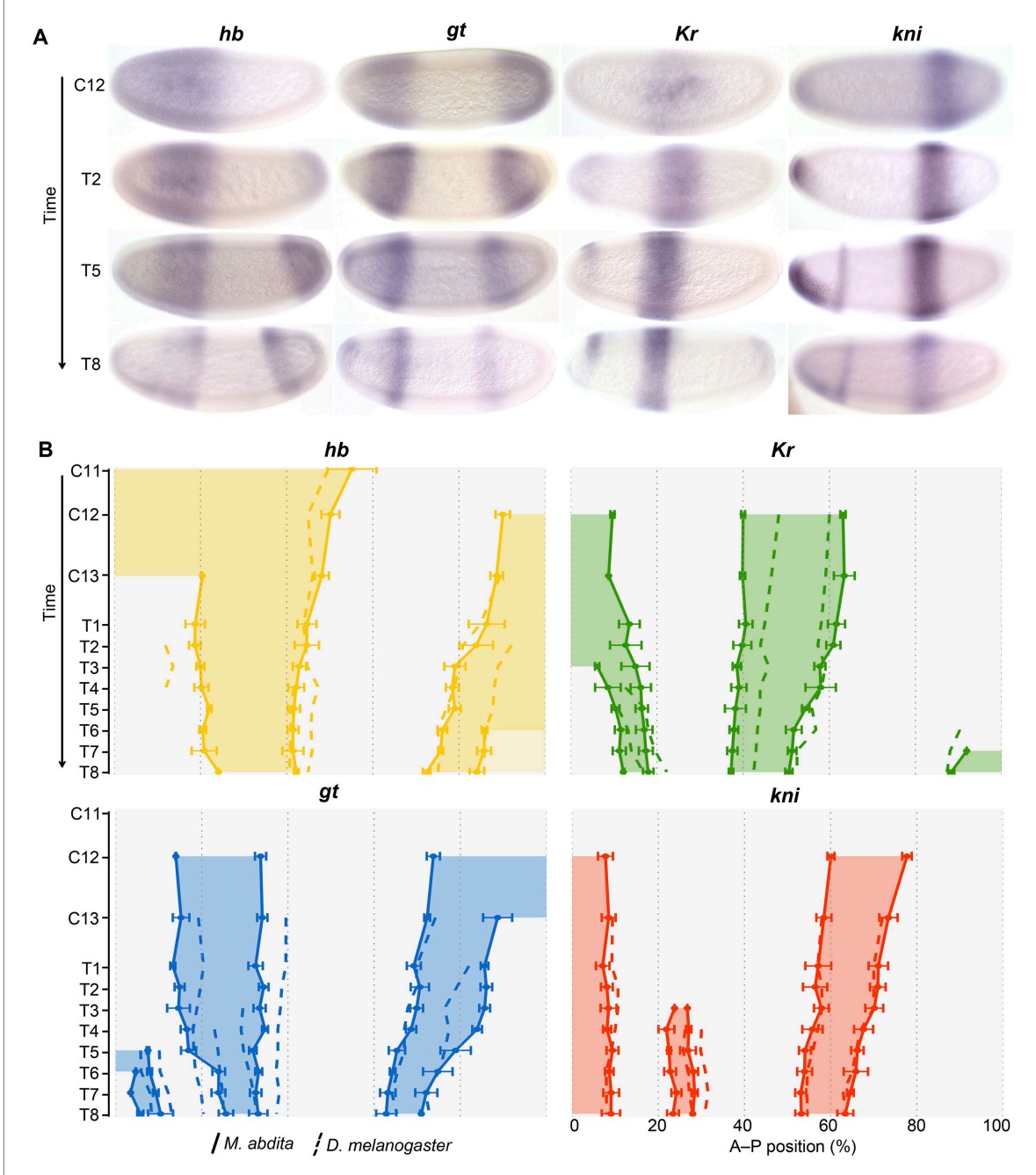

**Figure 3**. Trunk gap gene expression in *M. abdita*. (**A**) Wild-type mRNA expression patterns for trunk gap genes *hunchback (hb)*, *Krüppel (Kr), giant (gt)*, and *knirps (kni)* are shown in *M. abdita* embryos at selected time points (cleavage cycle 12, C12; and C14A, time classes T2/5/8). Embryos are shown in lateral view; anterior is to the left, dorsal up. (**B**) Space-time plots of gap gene expression in *M. abdita* (solid lines, filled areas) and *D. melanogaster* (dashed lines). *hb* is shown in yellow, *Kr* in green, *gt* in blue, *kni* in red. Lines represent positions of half-maximal expression for median boundaries; error bars for *M. abdita* data represent 1.5x median absolute deviation (MAD), as in *Figure 2B*. Lightly coloured *hb* in T6–T8 represents down-regulation. Time progresses downward in (**A**) and (**B**). C12/13: cleavage cycles 12/13; T1–8: time classes within C14A as defined in *Wotton et al. (2014)* for *M. abdita* and in *Surkova et al. (2008b)* for *D. melanogaster*.

more detail in *Supplementary file 4*. We monitor the sensitivity of expression dynamics to specific knock-downs at a temporal resolution equivalent to our wild-type dataset. This high temporal resolution, in combination with a large sample size, enables us to detect subtle alterations in gene expression dynamics that would have been missed by traditional, qualitative RNAi analyses. We structure our

**Box 1.** Maternal co-ordinate and gap gene mRNA expression in *M. abdita*.

Maternal co-ordinate genes: in *M. abdita*, anteriorly localised *bcd* transcripts extend about twice as far to the posterior, and disappear earlier, than in *D. melanogaster* (*Stauber et al., 2002*, *2000*, *1999*). There is no maternal *cad* expression (*Stauber et al., 2008*). Zygotic *cad* is detectable from C12 onwards: its anterior limit of expression is located further anterior, and its late posterior stripe is wider, than the homologous expression features in *D. melanogaster*.

Terminal gap genes: the posterior domains of the terminal gap genes *tll* and *hkb* are similar in both species, although *tll* is more restricted to the pole region in *M. abdita*.

*hb:* in contrast to earlier qualitative descriptions of *hb* expression in *M. abdita* (*Stauber et al., 2000*; *Lemke et al., 2008*), we detect significant differences in the dynamics of zygotic *hb* expression between the two species. The zygotic anterior *hb* domain forms around the same time in both species (C10/11) but extends further to the posterior in *M. abdita*. Its posterior boundary shifts to the anterior over time (*Figure 3A,B*). In contrast, it remains stationary throughout C14A in *D. melanogaster* (*Figure 3B*). The posterior *hb* domain appears during C12 in both species. This domain retracts from the posterior pole at T2 in *D. melanogaster*. Retraction is delayed and exhibits more complex dynamics in *M. abdita*. It starts with down-regulation of expression in the pole region from T6 onwards, and full retraction only occurs around T8 (*Figure 3B*).

*Kr:* the central *Kr* domain narrows over time in both species but remains wider in *M. abdita* than in *D. melanogaster* throughout the entire blastoderm stage (*Figure 3B*). Its anterior boundary is located further anterior and shifts less, while its posterior boundary shifts more, in *M. abdita* than in *D. melanogaster* (*Figure 3B*).

*gt:* the anterior *gt* domain is located somewhat further anterior in *M. abdita* compared to *D. melanogaster* (*Figure 3B*). Unlike *D. melanogaster*, the posterior boundary of the anterior *gt* domain does not shift in *M. abdita* (*Figure 3B*). Expression dynamics also differ in the posterior domain: although retraction from the posterior pole occurs at similar stages in both species, the shift of the posterior *gt* boundary is delayed in *M. abdita*; once it occurs, however, it is much faster than in *D. melanogaster* (*Figure 3B*).

*kni:* the abdominal domain of *kni* is the most similar of the trunk gap domains between the two species in terms of position, width, and the extent of its shift (*Figure 3B*).

discussion of interactions among gap genes in *M. abdita* according to the basic regulatory principles they contribute to in *D. melanogaster* (reviewed in *Jaeger, 2011*).

## Repression between complementary gap genes (alternating cushions)

In *D. melanogaster*, there are two pairs of gap genes whose expression domains are mutually exclusive: *hb* and *kni* as well as *Kr* and *gt.* Both of these pairs are positioned in a staggered overlapping arrangement along the A–P axis. We observe the same overall relative positioning of gap gene expression patterns in *M. abdita* (*Figure 3*). However, in this species mutually exclusive domains do show a tendency to overlap slightly, especially during the early blastoderm stage, a phenomenon never observed in *D. melanogaster*. The complementary patterns observed in *D. melanogaster* are established by strong mutual repression between *hb/kni* and *Kr/gt*, respectively (reviewed in *Jaeger, 2011*; see inset in *Figure 1*). We tested whether this is also the case in *M. abdita*.

In *kni* knock-down embryos (*Figure 6A–C*) we observe a posterior expansion of the anterior *hb* domain (in 30 out of 54 RNAi-treated embryos) and an anterior expansion of posterior *hb* (35/52). This phenotype is stronger than that of *kni* mutants in *D. melanogaster*, where only the anterior expansion of posterior *hb* has been observed but the anterior domain remains unaffected (*Jäckle et al., 1986*; *Houchmandzadeh et al., 2002*; *Clyde et al., 2003*). In *hb* knock-down embryos (*Figure 6D–F*), we observe an anterior expansion of the abdominal *kni* domain (9/14). A similar expansion is seen in maternal and zygotic *hb* null mutants of *D. melanogaster* but is considerably more pronounced than in *M. abdita* (*Hülskamp et al., 1990*; *Rothe et al., 1994*; *Clyde et al., 2003*). This could be due to the fact that *Kr* is not down-regulated in *M. abdita hb* RNAi (see below), while it is strongly reduced in *D. melanogaster hb* mutants (*Hülskamp et al., 1990*). As in *D. melanogaster hb* mutants, *kni* does not expand to the posterior in *M. abdita hb* knock-down embryos (*Figure 6D–F*). This could be due to

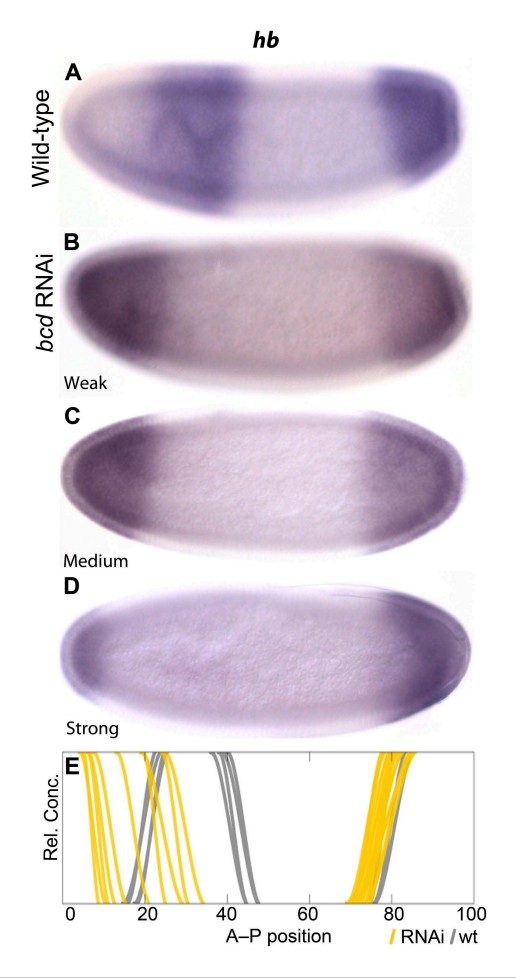

**Figure 4**. Gap domain boundary positioning is dependent on Bicoid levels in *M. abdita*. *hb* expression (purple) is shown in wild-type (**A**) and in *bcd* RNAi-treated embryos (**B–D**). The position of the posterior boundary of the anterior *hb* domain moves anteriorly as Bcd levels are reduced by RNAi. (**E**) Summary graph comparing wild-type boundary positions (grey) to boundary positions affected by RNAi (yellow lines). All embryos are at time class T4. Embryo images show lateral views: anterior is to the left, dorsal is up.

residual repression by *gt* and the terminal gap genes in this region (*Jaeger et al., 2004b*).

In *Kr* knock-down embryos (*Figure 6G–I*), we observe expansion of the anterior (17/20) and posterior (8/20) domains of *gt* into the central region of the embryo. This expansion is stronger in the anterior, where *gt* and *Kr* domains slightly overlap. A similar central expansion has been observed in *D. melanogaster Kr* mutants by some authors (*Mohler et al., 1989*; *Eldon and Pirrotta, 1991*; *Kraut and Levine, 1991a*; *Capovilla et al., 1992*), although a later quantitative analysis (*Surkova et al., 2013*) did not find any expansion of the anterior *gt* domain. Surprisingly, we also detect retraction defects of *gt* at the posterior pole region in some embryos (11/20). This effect is unlikely to be direct, since *Kr* is not expressed in this part of the embryo at the relevant stages of development (see *Figure 3B*). In *gt* knock-down embryos (*Figure 6J–L*), we do not observe any unambiguously detectable change in the boundary positions of *Kr*, although there may be a very subtle posterior displacement of its central domain (see *Supplementary file 4*). The effect of *gt* on *Kr* in *D. melanogaster* is also subtle and only visible at late stages (*Gaul and Jäckle, 1987*; *Reinitz and Levine, 1990*; *Eldon and Pirrotta, 1991*; *Strunk et al., 2001*).

In summary, our results indicate that mutual repression between *hb* and *kni* is conserved, but the strength of these repressive interactions differs considerably between *M. abdita* and *D. melanogaster*: we detect weaker inhibition of *kni* by *hb* and stronger inhibition of *hb* by *kni*. This repressive asymmetry is further supported by the alteration of *hb* expression following *kni* deregulation in *gt* RNAi embryos (see below), and may contribute to the shift of the posterior boundary of anterior *hb*, which is only observed in *M. abdita* but not in *D. melanogaster*. In addition, our results are consistent with the interpretation that the cross-repressive interactions between *Kr* and *gt* are very similar in both species, although over-expression studies as performed in *D. melanogaster* (*Eldon and Pirrotta, 1991*; *Kraut and Levine, 1991b*) will be required to definitely confirm repression of *Kr* by *gt* in *M. abdita*.

## Repression between overlapping gap genes (shift mechanism)

In *D. melanogaster,* cross-repression between neighbouring overlapping gap domains causes an anterior shift of expression boundaries over time (*Jaeger et al., 2004b*; *Crombach et al., 2012a*). These interactions show an asymmetrical bias towards the central *Kr* domain: posterior neighbours inhibit anterior ones, such that *Kr* is repressed by abdominal *kni*, *kni* by posterior *gt*, and *gt* by posterior *hb* (see inset in *Figure 1*). In contrast, repression from anterior to posterior neighbour is weak or completely absent in each case. We address each of these asymmetric feedbacks in turn (*Figure 7*).

The evidence for posterior-to-anterior neighbour repression is strong and unambiguous in *M. abdita*. In *hb* knock-down embryos (*Figure 7A–C*), posterior *gt* fails to retract from the posterior pole (24/35)

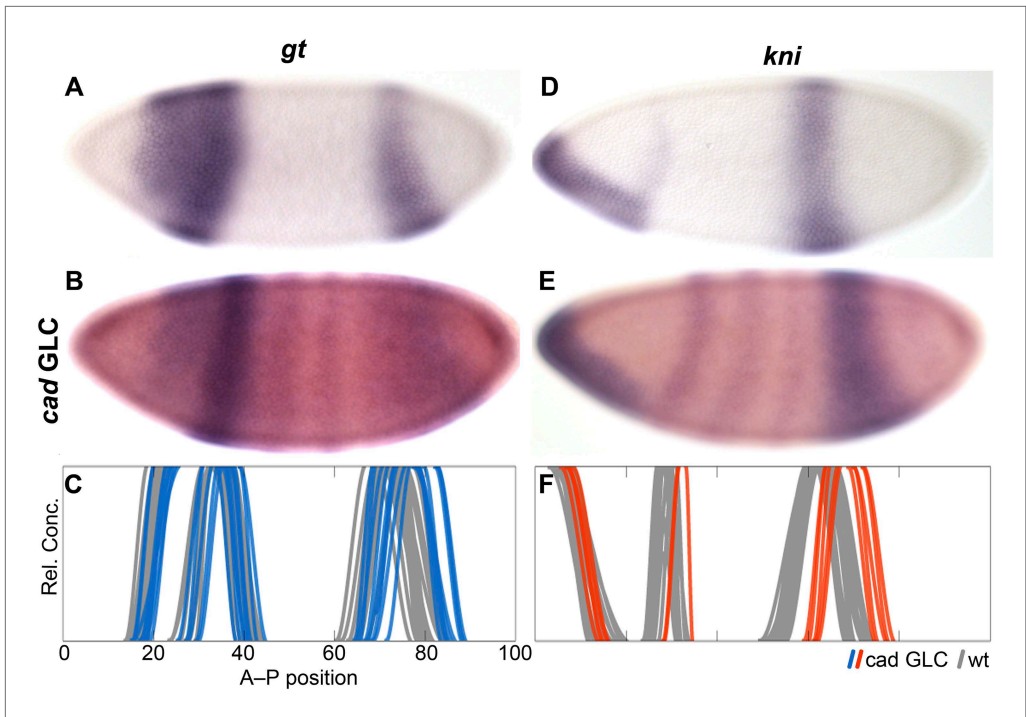

**Figure 5**. Absence of maternal cad delays posterior patterning in *D. melanogaster*. The expression of *gt* (**A–C**) and *hb* (**D–F**) is shown in wild-type embryos (**A**, **D**) and in *cad* germ-line clones (*cad* GLC) lacking only maternal *cad* (**B**, **E**; purple stain: *gt* or *kni*, red stain: *even-skipped*). (**C**, **F**) Summary graphs comparing wild-type boundary positions (grey) to boundary positions in *cad* GLC (coloured lines) show posterior domains of *gt* (**C**) and abdominal domains of *kni* (**F**) that are displaced towards the posterior. Embryos are at time class T3 (*gt*; **A–C**) and T4 (*kni*; **D–F**). Embryo images show lateral views: anterior is to the left, dorsal is up.

or does not retract fully (4/35). The same effect is seen in *D. melanogaster hb* mutants (**Mohler et al., 1989**; **Eldon and Pirrotta, 1991**). The anterior *gt* domain also shows increased variability upon *hb* knock-down (see **Supplementary file 4**). In *gt* knock-down embryos (**Figure 7D–F**), the abdominal *kni* domain expands posteriorly (11/25). A similar expansion was reported by some authors for *gt* mutants in *D. melanogaster* (**Eldon and Pirrotta, 1991**), but another study could not confirm this effect (**Rothe et al., 1994**). In *kni* knock-down embryos (**Figure 7G–I**), the central *Kr* domain shows a very pronounced expansion to the posterior (26/40). The same phenotype has been observed in *kni* mutants of *D. melanogaster* (**Jäckle et al., 1986**; **Gaul and Jäckle, 1987**; **Harding and Levine, 1988**;

**Table 2.** Overview of the RNAi dataset for *M. abdita*

|  | *hb* RNAi | *gt* RNAi | *Kr* RNAi | *kni* RNAi | *tll* RNAi | *hkb* RNAi | *tll:hkb* RNAi |
|---|---|---|---|---|---|---|---|
| *hb* | n/a | 17/28 (60%) | 21/41 (51%) | 35/52 (67%) | 12/29 (41%) | 21/22 (95%) | 15/15 (100%) |
| *gt* | 28/35 (80%) | n/a | 13/20 (65%) | 24/30 (80%) | 24/30 (80%) | 10/10 (100%) | 35/36 (97%) |
| *Kr* | 24/53 (46%) | 7/17 (41%) | n/a | 26/40 (65%) | 7/18 (38%) | 10/10 (100%) | 12/21 (57%) |
| *kni* | 9/14 (64%) | 11/25 (44%) | 3/21 (14%) | n/a | 6/44 (14%) | 4/10 (40%) | 16/16 (100%) |
| Total | 102 | 70 | 82 | 122 | 121 | 52 | 88 |

A total of 637 RNAi-treated embryos were used for the analysis. This table shows the number of embryos that were stained for each of the trunk gap genes (rows), in each RNAi-treated background (columns), and the number of embryos that showed a knock-down phenotype (see also percentages). The total number of embryos used for each knock-down experiment is shown in the bottom row. A more detailed breakdown of embryos per cleavage cycle and time class, including detailed plots of boundary positions, is provided in **Supplementary file 4**.

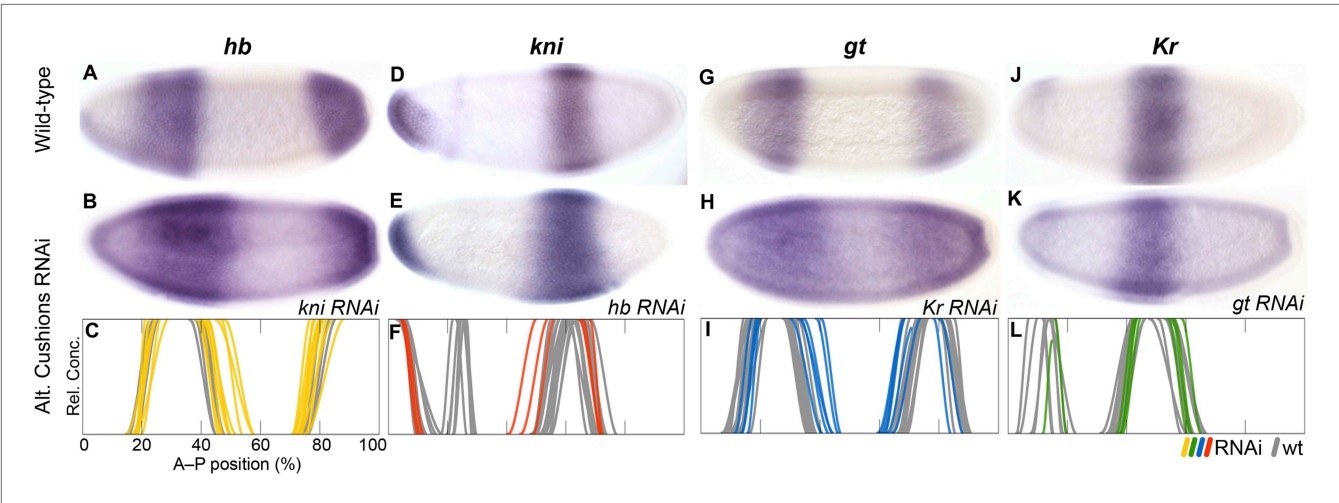

**Figure 6**. RNAi knock-down of complementary gap genes (alternating cushions). Columns show the expression of *hb* (**A–C**; yellow), *kni* (**D–F**; red), *gt* (**G–I**; blue), and *Kr* (**J–L**; green) in wild-type embryos (top row; **A, D, G, J**), in RNAi-treated embryos as indicated (middle row; **B, E, H, K**), and as summary graphs comparing wild-type boundary positions (grey) to boundary positions affected by RNAi (coloured lines) (bottom row; **C, F, I, L**). All embryos are at time class T4. Embryo images show lateral views: anterior is to the left, dorsal is up.

*Strunk et al., 2001*), although a more recent quantitative study was not able to detect it (*Surkova et al., 2013*).

In *M. abdita,* the posterior boundary of the anterior *hb* domain also shifts to the anterior, while it remains stationary in *D. melanogaster* (*Figure 3B*). Accordingly, the effect of *Kr* on *hb* is stronger in *M. abdita* than in *D. melanogaster*. The anterior *hb* domain expands posteriorly in *Kr* knock-down embryos (21/41) (*Figure 7J–L*). In addition, both the anterior and posterior domains of *hb* show increased variability compared to the wild-type (see *Supplementary file 4*). Both of these effects are much weaker in *D. melanogaster*, where some authors have reported a slight posterior displacement of the *hb* boundary (*Jäckle et al., 1986*; *Hülskamp et al., 1990*; *Wu et al., 1998*; *Clyde et al., 2003*), but more recent quantitative analyses have neither found an effect on boundary position nor variability (*Houchmandzadeh et al., 2002*; *Surkova et al., 2013*). In *hb* knock-down embryos, the central *Kr* domain expands towards the anterior (24/52; *Figure 7Q,R*). A very similar effect is seen in *D. melanogaster hb* mutants (*Jäckle et al., 1986*; *Gaul and Jäckle, 1987*; *Harding and Levine, 1988*; *Hülskamp et al., 1990*).

In contrast, we do not find any evidence for anterior-to-posterior neighbour repression in *M. abdita*, and those effects we do detect are likely to be indirect. In *gt* knock-down embryos, posterior *hb* is slightly displaced towards the posterior (17/28) (Figure 6S, T), suggesting weak activation rather than repression of *hb* by *gt*. However, this effect may be indirect, mediated by the expansion of the abdominal *kni* domain in *hb* knock-downs (*Figure 7E,F*). The posterior *hb* domain is not affected at all in *D. melanogaster gt* mutants (*Eldon and Pirrotta, 1991*; *Strunk et al., 2001*). Similarly, in *kni* RNAi embryos, the anterior boundary of the posterior *gt* domain is slightly displaced towards the posterior (24/30), while the posterior boundary of this domain fails to retract from the pole (18/30) or does not fully retract (6/30). Again, this effect may be indirect through deregulation of *hb* and *Kr* (*Figure 5K,L*; Figure 6S,T). In *D. melanogaster kni* mutants, expression levels in the posterior *gt* domain are reduced (*Eldon and Pirrotta, 1991*; *Kraut and Levine, 1991a*), and there are slight defects in the posterior boundary (*Mohler et al., 1989*; *Eldon and Pirrotta, 1991*). Just as in *M. abdita*, these effects may be indirect, since *kni* is not expressed in the affected regions (*Jaeger, 2011*). Finally, we did not observe any effects on *kni* in *Kr* knock-down embryos (*Figure 7O,P*). In particular, we do not detect any evident reduction in expression levels in the abdominal *kni* domain, as observed in *D. melanogaster Kr* mutants (*Pankratz et al., 1989*; *Surkova et al., 2013*).

In summary, our results indicate that there is a strong asymmetry towards posterior-to-anterior neighbour repression in *M. abdita*. The less ambiguous nature of the evidence compared to *D. melanogaster* suggests that interactions involved in gap domain shifts are generally stronger in this species.

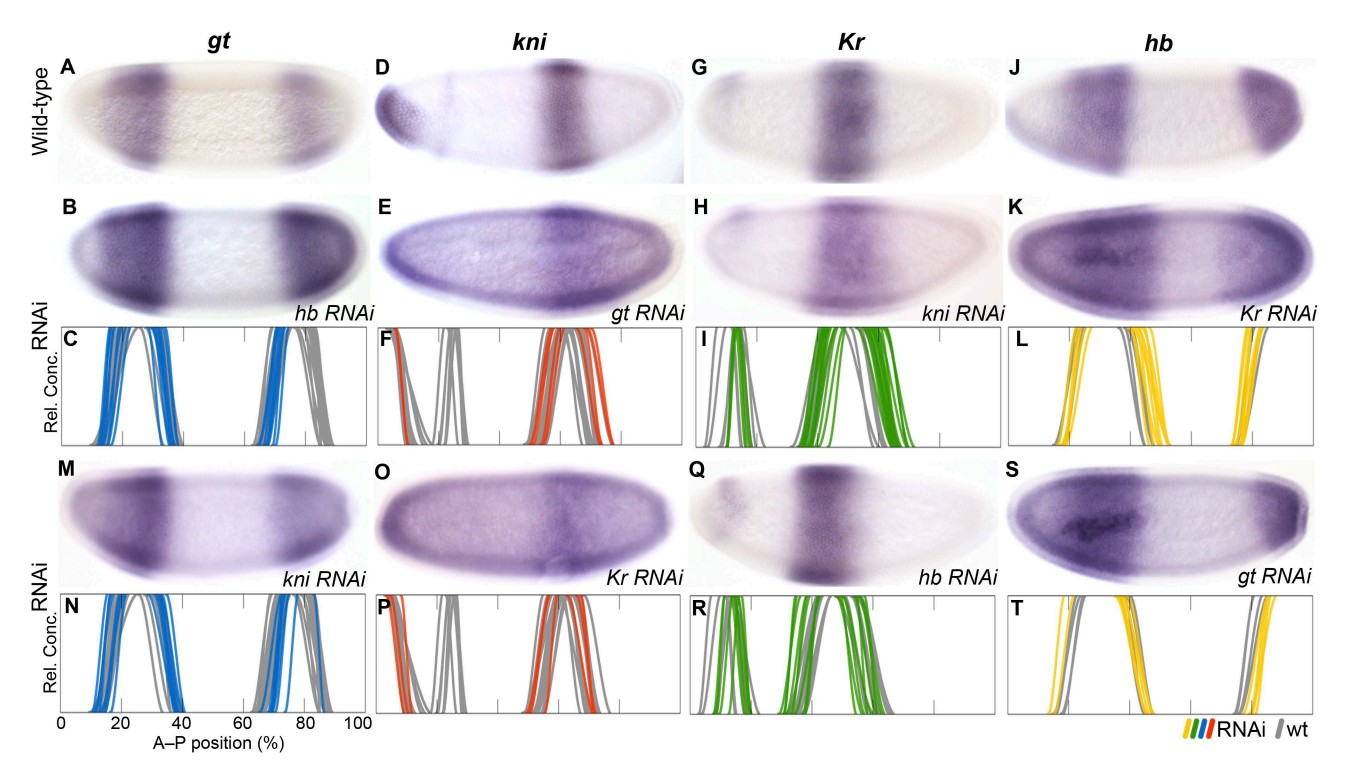

**Figure 7**. RNAi knock-down of overlapping gap genes (shift mechanisms and others). Columns show the expression of *gt* (**A–C**, **M**, **N**; blue), *kni* (**D–F**, **O**, **P**; red), *Kr* (**G–I**, **Q**, **R**; green), and *hb* (**J–L**, **S**, **T**; yellow) in wild-type embryos (top row; **A**, **D**, **G**, **J**), in RNAi-treated embryos as indicated (rows 2 and 4; **B**, **E**, **H**, **K**; **M**, **O**, **Q**, **S**), and as summary graphs comparing wild-type boundary positions (grey) to boundary positions affected by RNAi (coloured lines) (rows 3 and 5; **C**, **F**, **I**, **L**; **N**, **P**, **R**, **T**). All embryos are at time class T4. Embryo images show lateral views: anterior is to the left, dorsal is up.

The observed difference in the repressive balance between *Kr* and *hb*, in particular, is essential to explain the altered dynamics of boundary positioning between the two species (see Conclusions).

## Repression by terminal gap genes

The terminal gap genes *tll* and *hkb* are responsible for repression of trunk gap genes in the posterior terminal region of the *D. melanogaster* blastoderm embryo (see grey inset in *Figure 1*). Most of this terminal repression is carried out by *tll*, while the only known non-redundant role of *hkb* in gap gene patterning consists of positioning the posterior boundary of the posterior *hb* domain (*Ashyraliyev et al., 2009*; *Jaeger, 2011*). In *M. abdita*, *tll* and *hkb* play similar roles, although the amount of their respective regulatory contributions is altered compared to *D. melanogaster* (*Figure 8*).

In *tll* knock-down embryos, we find that the posterior *hb* domain is reduced in size (12/29) (*Figure 8A2–3*), absent at early blastoderm stages (C13-T5; 4/17), and fails to retract from the posterior pole at later times (T7; 1/1). The anterior boundary of the posterior *gt* domain is displaced towards the posterior in some embryos (13/30), while it fails to retract completely (15/30) or does not retract fully (9/30) in others (*Figure 8D2–3*). Abdominal *kni* shows no, or very weak, posterior displacement of both of its boundaries (6/44) (*Figure 8C2–3*). Only subtle effects are seen on *Kr* (*Figure 8B2–3*; see also *Supplementary file 4*). In *D. melanogaster tll* mutants, posterior *hb* is strongly reduced or absent (*Casanova, 1990*; *Reinitz and Levine, 1990*; *Brönner and Jäckle, 1991*), posterior *gt* retraction is delayed (*Brönner and Jäckle, 1991*; *Eldon and Pirrotta, 1991*; *Kraut and Levine, 1991a*), abdominal *kni* expands to the posterior (*Pankratz et al., 1989*; *Brönner and Jäckle, 1991*; *Rothe et al., 1994*), and *Kr* is not affected (*Gaul and Jäckle, 1987*; *Reinitz and Levine, 1990*).

In *hkb* knock-down embryos, we find that posterior *hb* is reduced in size (21/22) (*Figure 8A4–5*), fails to retract fully from the posterior pole during T6–T8 (2/11), or shows no reduction of expression in the posterior pole region at all (9/11). The posterior *gt* domain is also reduced in size (8/10) and fails

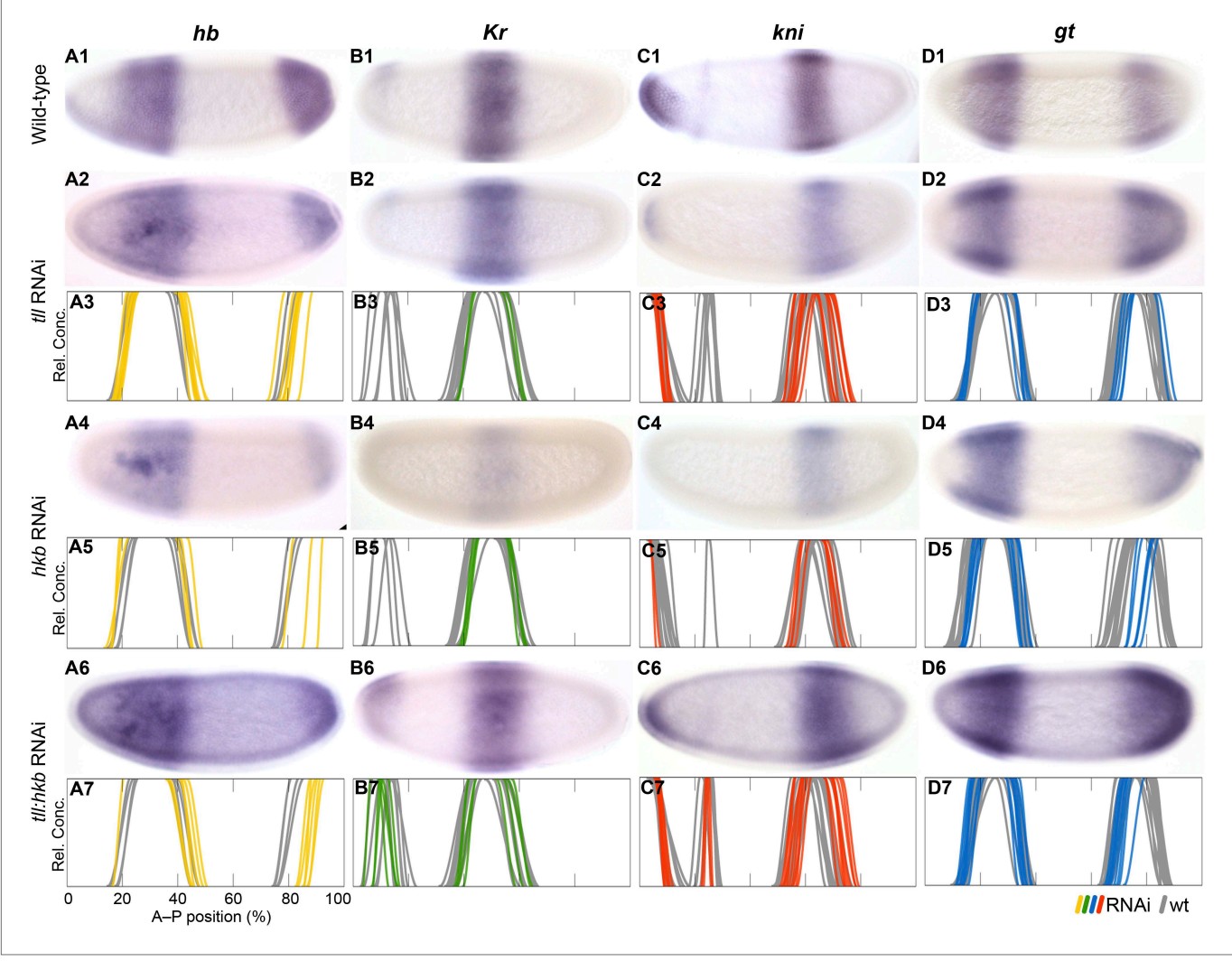

**Figure 8**. Knock-down of the terminal gap genes. Columns show the expression of *hb* (**A1**–**A7**; yellow), *Kr* (**B1**–**B7**; green), *kni* (**C1**–**C7**; red), *gt* (**D1**–**D7**; blue) in wild-type embryos (top row; **A1**, **B1**, **C1**, **D1**), in RNAi-treated embryos as indicated (rows 2, 4, and 6; **A**–**D**), and as summary graphs comparing wild-type boundary positions (grey) to boundary positions affected by RNAi (coloured lines) (rows 3, 5, and 7; **A**–**D**). All embryos are at time class T4. Embryo images show lateral views: anterior is to the left, dorsal is up.

to retract (10/10) (*Figure 8D4–5*). Abdominal *kni* shows no, or very weak, posterior displacement of the anterior boundary (4/10) (*Figure 8C4–5*). No obvious effect is seen on the position of *Kr* domain boundaries (*Figure 7B4–5*). In *D. melanogaster hkb* mutants, the posterior *hb* domain fails to retract from the posterior pole (*Casanova, 1990*; *Brönner and Jäckle, 1991*), *hkb* represses *gt* when over-expressed (*gt* is unaltered in *hkb* mutants) (*Brönner et al., 1994*), and *Kr* expands further in a *tll hkb* compared to *tll* background when combined with a posterior system mutant (*Weigel et al., 1990*). The effect of *hkb* on *kni* remains unknown.

To further test the synergy between *tll* and *hkb* in *M. abdita*, we performed a double knock-down experiment. In *tll hkb* RNAi embryos, we find that posterior *hb* is strongly reduced in size (15/15) (*Figure 8A6–7*) and either completely fails to retract (3/6) or fails to retract fully (3/6) during T6 to T8. The posterior *gt* domain is also reduced (21/36) and fails to retract (33/36) or does not retract fully (2/36) (*Figure 8D6–7*). Abdominal *kni* shows posterior displacement (16/16) (*Figure 8C6–7*), and a very subtle posterior shift is seen for central *Kr* (12/21) (*Figure 8B6–7*). In *D. melanogaster tll hkb* double mutants, posterior *hb* is absent or strongly reduced (*Casanova, 1990*), posterior *gt* fails to retract (*Brönner and Jäckle, 1991*), abdominal *kni* expands to the posterior (*Brönner and Jäckle, 1991*), and so does *Kr* if combined with a posterior system mutant background (*Weigel et al., 1990*).

Taken together, our results indicate that the regulatory effect of *tll* and *hkb* on trunk gap gene expression is similar in both species, although *hkb* may have a more prominent, and less redundant, role in regulating trunk gap genes in *M. abdita* than in *D. melanogaster*, as indicated by the synergistic effect on *Kr* and *kni* expression observed in *tll hkb* double knock-downs.

## Conclusions

In this paper, we present a quantitative and systematic comparative analysis of the expression dynamics and regulatory structure of a developmental gene regulatory network—the gap gene system—responsible for A–P patterning in early dipteran embryos. Through the use of a medium-throughput data quantification pipeline (*Crombach et al., 2012b*) and a recently developed detailed embryonic staging scheme (*Wotton et al., 2014*), we are able to characterise and compare gap gene expression dynamics between *M. abdita* and *D. melanogaster* with unprecedented accuracy and spatio-temporal resolution (*Figure 3*). Our fine-grained analysis reveals that, on the one hand, expression dynamics differ significantly between the two species. For instance, the posterior boundary of the anterior *hb* domain shifts to the anterior over time in *M. abdita*, while it remains stationary throughout the blastoderm stage in *D. melanogaster*. On the other hand, the output of the system is strongly conserved across 180 million years of evolution. The position and relative arrangement of gap domains converge in both species towards the onset of gastrulation. Each of these domains is initially set up more posteriorly, but 'catches up' with its equivalent in *D. melanogaster* through more pronounced anterior boundary shifts in *M. abdita* (*Figure 9A*). The timing of this convergence is delayed in the posterior of the embryo: posterior domain boundaries of posterior *hb* start to coincide at T8 (when this border becomes fully resolved in *M. abdita*), those of *gt* around T6, those of abdominal *kni* at T3, those of central *Kr* at T2, and those of anterior *hb* already around T1 (although the latter diverge again temporarily at later stages) (*Figure 9A*).

These differences in expression dynamics are a consequence of the altered distribution of maternal co-ordinate gene products in *M. abdita*. Posterior gap domains appear further posterior—due to the

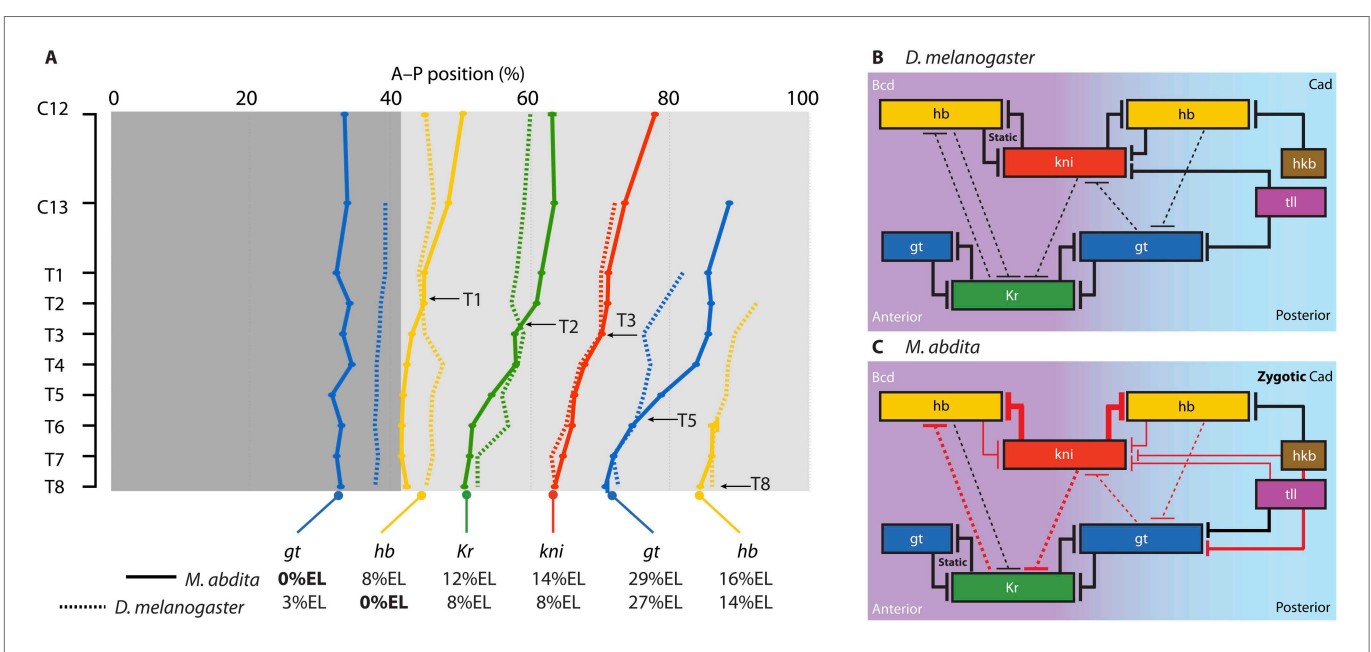

**Figure 9**. Comparison of posterior boundary positions and gene network structure between *M. abdita* and *D. melanogaster*. (**A**) This graph shows the position of the posterior boundaries of anterior *gt* (blue) and *hb* (yellow), central *Kr* (green), abdominal *kni* (red), and posterior *gt* (blue) and *hb* (yellow). Initial boundary positions are either more anterior (dark grey) or more posterior (light grey background) in *M. abdita* (solid) than in *D. melanogaster* (dotted lines). Arrows with time classes T1–T8 indicate when boundary positions first converge between the two species. Time flows downward. (**B** and **C**) Gap gene network structure for *D. melanogaster* (**B**) and *M. abdita* (**C**) as reconstructed from our knock-down analysis, displayed using the same layout as the inset of *Figure 1*. Red T-bar connectors indicate differences in interaction strength (indicated by line width) between *M. abdita* and *D. melanogaster*. See text for details.

broadened distribution and influence of *bcd*—and shift to the anterior later than in *D. melanogaster*—due to the absence of maternal *cad*. This is later compensated by pronounced, faster domain shifts in the second half of C14A, an effect most clearly exhibited by *Kr* and *gt* (**Figure 9A**). Correspondingly, our knock-down analysis of the gap gene network in *M. abdita* reveals a stronger posterior-to-anterior bias in cross-repression between neighbouring domains (**Figure 9**; compare B to C). An equivalent posterior bias is responsible for the anterior shift in the posterior boundary of anterior *hb.* In this case, it is not only repression of *hb* by Kr, but also repression by Kni, which is markedly stronger than the corresponding reverse interactions (**Figure 9B,C**). Taken together, this suggests that the more pronounced domain shifts of *hb* and posterior boundaries in *M. abdita* are caused by increased posterior cross-regulatory bias.

Our analysis reveals how different developmental dynamics and regulatory interaction strengths in the gap gene network produce extremely similar patterns at the onset of gastrulation, despite significant differences in the initial phases of maternal co-ordinate and gap gene expression. This constitutes developmental system drift of a quantitative kind: instead of changes in the qualitative nature of interactions (e.g., a switch from activation to repression), we observe neutral or compensatory evolution through tuning of regulatory interaction strengths. While most published examples of developmental system drift are of the first, qualitative, kind, there is one previous study that proposed quantitative changes in signalling strength to account for the evolution of the vulval patterning system in *Caenorhabditis elegans* (**Hoyos et al., 2011**). This analysis focuses on the relative contributions of different signalling modes—long-range gradient vs juxtacrine signalling relay—to downstream cellular patterning. In contrast, we have investigated the regulatory mechanisms of compensatory network evolution in full genetic detail, at the level of individual regulatory interactions.

The precise molecular mechanism for such changes remains unknown in *M. abdita* at this point—and will be subject to future studies. It is likely to be based on altered numbers, affinities, or arrangements of transcription factor binding sites in cis-regulatory elements (**Levine, 2010**; **Wittkopp and Kalay, 2011**) as observed in other dipteran lineages (**Ludwig and Kreitman, 1995**; **Bonneton et al., 1997**; **Ludwig et al., 1998, 2000, 2005**; **Hancock et al., 1999**; **McGregor et al., 2001**; **Shaw et al., 2002**; **Gompel et al., 2005**; **Hare et al., 2008**; **Jeong et al., 2008**; **Williams et al., 2008**; **Peterson et al., 2009**; **Bradley et al., 2010**; **Frankel et al., 2011**; **Wunderlich et al., 2012**; **Paris et al., 2013**). Although many such changes at the sequence level do not seem to significantly affect levels of gene expression, there are others that do (**Wunderlich et al., 2012**; **Arnold et al., 2014**). Here, we show that even these changes—that are not neutral at the level of cis-regulatory elements but lead to a quantitative difference in interaction strength—can lead to neutral drift if we consider their phenotypic effects at the systems- or network-level. Based on the observed high turnover rates for the evolution of binding sites in various evolutionary lineages (**McGregor et al., 2001**; **Dermitzakis and Clark, 2002**; **Ludwig, 2002**; **Costas et al., 2003**; **Dermitzakis et al., 2003**; **Moses et al., 2006**; **Yáñez-Cuna et al., 2013**), we believe that such quantitative system drift is a common feature of network evolution. It supplies a mechanism for exploring alternative genotype spaces, increasing the range of mutationally accessible novel phenotypes (**Wagner, 2011**; **Jaeger and Monk, 2014**), and allows for dynamically different alternative regulatory processes to evolve in specific developmental contexts, such as that of segmentation (**Oates et al., 2012**; **Richmond and Oates, 2012**; **Sarrazin et al., 2012**; **Jaeger and Sharpe, 2014**). Based on these considerations, quantitative system drift is likely to be important, not only for neutral phenotypic evolution but also for the de novo evolution of biological form.

Finally, quantitative system drift in early embryonic patterning networks, as reported here for the cyclorrhaphan gap gene system, provides an explanation for the developmental hourglass model (**Seidel, 1960**; **Sander, 1983**; **Slack et al., 1993**; **Duboule, 1994**; **Raff, 1996**). This model predicts a minimum of morphological divergence during intermediate embryonic stages. It has been confirmed at the level of gene expression in three recent studies (**Domazet-Lošo and Tautz, 2010**; **Kalinka et al., 2010**; **Quint et al., 2012**). We observe the same phenomenon in our explicitly spatio-temporal context: maternal co-ordinate and early gap gene expression differ significantly between species, but later converge to a common pattern (**Figure 9A**). Our knock-down analysis of the gap gene network in *M. abdita* provides causal regulatory explanations of how this can be achieved at the gene network-level.

## Materials and methods

*M. abdita* fly culture and embryo collection/fixation were carried out as described in (**Rafiqi, at al. 2011b**; **Rafiqi et al., 2011c**). Blastoderm stage embryos of *M. abdita*—corresponding to embryonic

stages 4 and 5 (*Campos-Ortega and Hartenstein, 1997*; *Wotton et al., 2014*)—were collected 4 hr (hrs) after egg laying. We visualised maternal-coordinate and gap gene mRNA expression patterns using an enzymatic (colorimetric) in situ hybridisation protocol as previously described (*Crombach et al., 2012a*). We used single probes for the majority of stains. Embryos double-stained for gap genes or for a gap gene along with the pair-rule gene *even-skipped (eve)* were used to confirm the position of expression domains relative to each other (see http://superfly.crg.eu for our full dataset; *Cicin-Sain et al., 2015*). We took four images of each stained embryo (*Crombach et al., 2012b*): a bright-field image from which the expression profile was extracted, a DIC image used to create the whole-embryo mask, and two images of the fluorescently counterstained nuclei and membrane morphology respectively, which were used to determine the time class of each embryo based on the staging scheme described in *Wotton et al. (2014)*. These images were processed to extract the position of expression domain boundaries as described in *Crombach et al. (2012b)* and shown in *Figure 2A,B*.

RNAi treatment was carried out as described in *Stauber et al. (2000)*; *Lemke et al. (2008)*; *Rafiqi et al. (2011a)*. Two distinct dsRNA constructs were injected per gene, one being the knock-down agent, the other one—with conserved domains excluded—serving as a control. Full length *bcd* (CAB40892.1), *cad* (ABY25304.1), *hb* (CAC00483.1), *Kr* (ABX89307.1), and *eve* (AAT92572.1) open reading frames (ORFs) were amplified from cDNA using primers based on published sequences (*Stauber et al., 1999*, *2000*, *2008*; *Bullock et al., 2004*; *Rafiqi et al., 2008*). *tll*, (KM489059), *gt* (KM489058), and *kni* (KM489060) (ORFs) were cloned from cDNA using degenerate and RACE PCR. *hkb* (KM489061) was cloned using sequence data from our published early embryonic transcriptome (http://diptex.crg.es; gene ID: comp8024) (*Jiménez-Guri et al., 2013*). *cad* germ-line clone females were generated as described in *Wu and Lengyel, (1998)* using fly stocks cad(2L-264-12-3)FRT40A/ CyO,hs-hid, w*; P{w⁺ ovo^D1} P{neo FRT}40A/T(1,2)OR64/SM6a, and y*w*,P{hs-Flp}122; If/CyO,hs-hid (kind gift from Uwe Irion). These females were mated to wild-type males to obtain embryos, all of which lack maternal *cad* activity but carry one zygotically active wild-type *cad* gene.

## Appendix I: *M. abdita* segmentation gene expression

In this section, we describe the quantified wild-type mRNA expression patterns of *M. abdita* maternal co-ordinate and gap genes in detail and compare them to the published literature in *D. melanogaster*.

## Maternal co-ordinate genes

### bicoid (bcd)

Our data confirm earlier reports that *bcd* transcripts are distributed more broadly in *M. abdita* than in *D. melanogaster* (*Stauber et al., 2002*, *2000*, *1999*). At C12, *bcd* transcripts extend from the anterior pole to 23% A–P position (*Figure 10*). This is 9–12%EL further to the posterior, or almost twice a far, as in *D. melanogaster* embryos at any time during the blastoderm stage (*Berleth et al., 1988*; *Little et al., 2011*; *Crombach et al., 2012b*). At subsequent stages, the distribution of *bcd* transcripts becomes more restricted and its posterior boundary moves to 16% A–P position at C13 and to 12% at C14A-T1/T2 (*Figure 10*). In comparison, we do not detect transcripts in *D. melanogaster* beyond 11% A–P position during C14A, in rough agreement with previous studies that reported a posterior boundary at around 14% (*Berleth et al., 1988*; *Little et al., 2011*). *bcd* transcripts in *M. abdita* disappear at C14A-T2, earlier than in *D. melanogaster* where they can still be detected up to C14A-T4 (*Figure 10*) (*Crombach et al., 2012a*).

### caudal (cad)

Unlike *D. melanogaster*, *M. abdita* does not show any maternal *cad* expression (*Stauber et al., 2008*). We first detect zygotic *cad* transcripts at C12 extending from around 34% A–P position to the posterior pole (*Figure 10*). At C13, the anterior boundary of *cad* expression is at a similar position in *M. abdita* and *D. melanogaster* (31 vs 33% A–P position; *Figure 10*) (*Crombach et al., 2012a*). However, at subsequent stages, *cad* retracts towards the posterior in *D. melanogaster* (39–42% A–P position during C14A-T2 to T5), while it maintains a stationary boundary at around 31–32% A–P position in *M. abdita* (until around C14A-T5; *Figure 10*; see also *Stauber et al., 2008*). At C14A-T5, *cad* transcripts start to disappear in most of the central region of the embryo in both species (*Figure 10*) (*Mlodzik and Gehring, 1987*; *Schulz and Tautz, 1995*). Around the same time, *cad* also retracts from the posterior pole forming a posterior stripe at 79–90% A–P position in *M. abdita* (*Figure 10*)

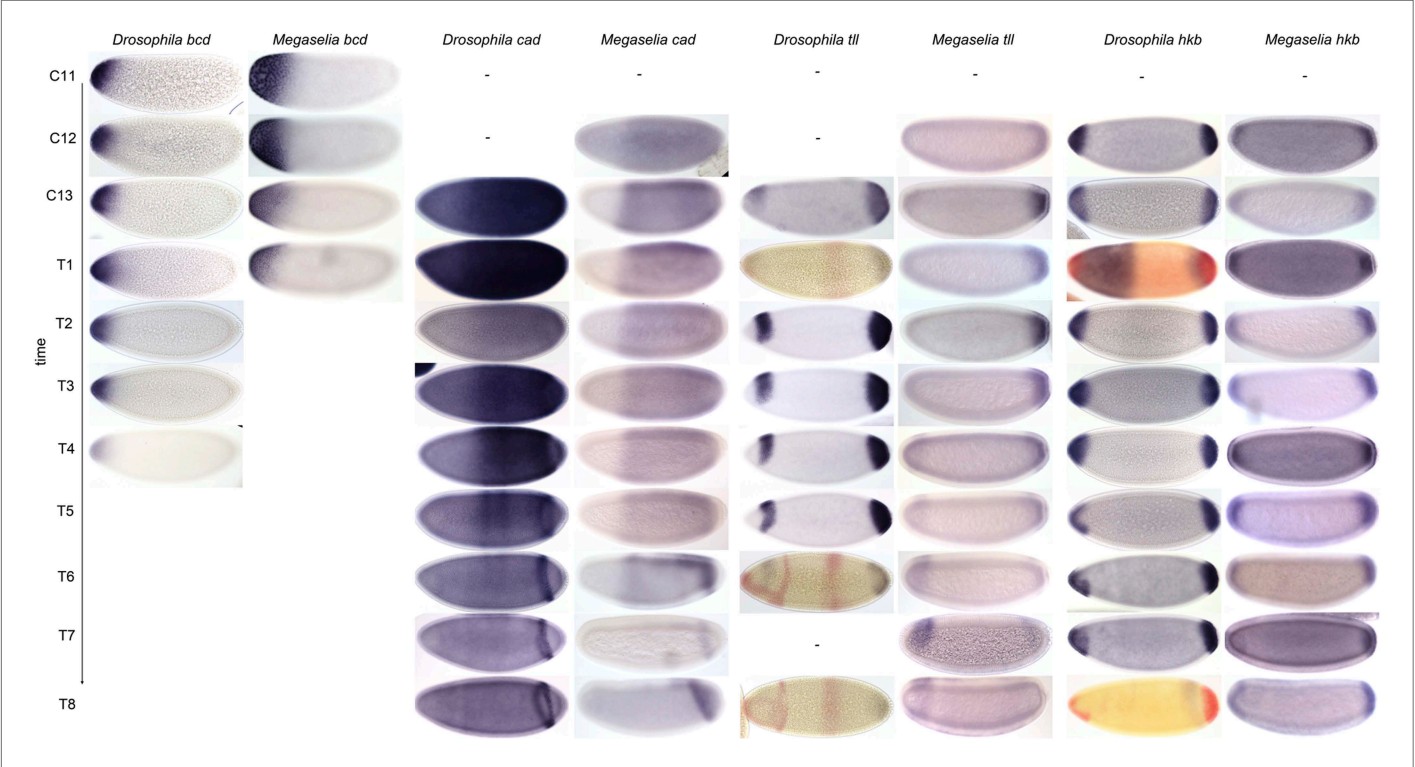

**Figure 10**. Expression data—maternal co-ordinate and terminal gap genes. Time series of mRNA expression patterns of the maternal genes *bcd* and *cad,* as well as the terminal gap genes *hkb* and *tll* are shown for *D. melanogaster* (left) and *M. abdita* (right). Expression patterns are visualised by colorimetric (enzymatic) in situ hybridisation using wide-field microscopy as described in *Crombach et al. (2012a)*. Relevant expression patterns are shown in blue, except for *hkb* at T1 and T8, which is shown in red. Red stain in *tll* (T1/6/8) shows *kni*. Single stains are shown for all other genes and time classes. Early stages at the top, time progresses downwards. Embryo images show lateral views: anterior to the left, dorsal up. C11–13: cleavage cycles 11–13; T1–8: time classes subdividing C14A as defined in *Surkova et al. (2008b)* and *Wotton et al. (2014)*. *D. melanogaster* patterns were published in *Crombach et al. (2012a)* and are shown for comparison.

(*Stauber et al., 2008*). This domain is much wider than its equivalent in *D. melanogaster* (about 11 vs 4% EL) (*Mlodzik and Gehring, 1987*; *Moreno and Morata, 1999*; *Crombach et al., 2012a*). It remains larger despite the posterior boundary shifting to the anterior at C14A-T8 causing its width to shrink to about 7% (vs. 4% EL in *D. melanogaster*; *Figure 10*) (*Crombach et al., 2012a*).

## Terminal gap genes
### tailless (tll)

We first detect *tll* expression at C12 in a domain at the posterior pole extending to 90% A–P position (*Figure 10*). This boundary remains more or less stationary up to C14A-T6, after which posterior *tll* expression appears to decrease. Despite beginning as a slightly larger domain at C13 than its equivalent in *D. melanogaster*, the posterior *tll* domain narrows and comes to occupy a slightly smaller region of the embryo (boundary at 84 vs 89% A–P position; *Figure 10*) (*Pignoni et al., 1990*; *Crombach et al., 2012a*). Expression dynamics in the anterior differ more markedly: while the anterior *tll* domain in *D. melanogaster* appears in C13, its equivalent in *M. abdita* does not appear until around C14-T4 (*Figure 10*) (*Pignoni et al., 1990*; *Crombach et al., 2012a*). At later stages, this domain retracts from the anterior pole and expands slightly to the posterior in both species.

### huckebein (hkb)

We first detect *hkb* expression at C12 in two domains, one at the anterior pole extending to 8% A–P position, and one at the posterior pole extending to 90% A–P position (*Figure 10*). The boundary of the anterior domain remains at 7–8% A–P position through C13 and most of C14A,

narrowing to 6% at C14A-T8. The extent of the posterior domain reaches up to around 89–90% A–P position between C14A-T1 and T6. In contrast to *D. melanogaster*, this domain narrows slightly during C14A-T7 and T8 to 92% A–P position. Other than that, expression dynamics and boundary positions are very similar to those reported for *D. melanogaster* (*Figure 10*) (*Brönner and Jäckle, 1996*; *Crombach et al., 2012a*).

## Trunk gap genes

### hunchback (hb)

Maternal *hb* transcripts have been reported to be uniformly distributed in the pre-blastoderm embryo of *M. abdita* (*Stauber et al., 2000*). We have not examined this maternal pattern any further. Earlier qualitative studies reported that zygotic expression of *hb* is similar in *M. abdita* and *D. melanogaster* (*Stauber et al., 2000*; *Lemke et al., 2008*). In contrast to these reports, we detect significant differences in expression dynamics between the two species. We first detect the formation of an anterior domain of zygotic *hb* expression in *M. abdita* at C11. This domain extends from 0 to 56% A–P position, reaching significantly further posterior than in *D. melanogaster*, where it is located from 0 to 50% A–P position (*Figure 11*) (*Tautz, 1988*; *Crauk and Dostatni, 2005*). In *M. abdita*, the posterior boundary of the anterior *hb* domain steadily shifts to the anterior over time (*Figure 11*). This shift covers 8% EL by C14A-T8. In contrast, the equivalent boundary in *D. melanogaster* remains stationary through C14A (see *Figure 3*) (*Surkova et al., 2008a*; *Crombach et al., 2012a*). By C13, some *M. abdita* embryos show a decrease of expression levels at the anterior pole (*Figure 11*). This process continues through C14A and results in the formation of a broad anterior stripe clearly visible at C14A-T4 (*Figure 11*). A slightly more intense stripe within the broad anterior *hb* domain in *M. abdita* may correspond to the parasegment 4 (PS4) stripe in *D. melanogaster* (*Tautz et al., 1987*; *Crombach et al., 2012a*).

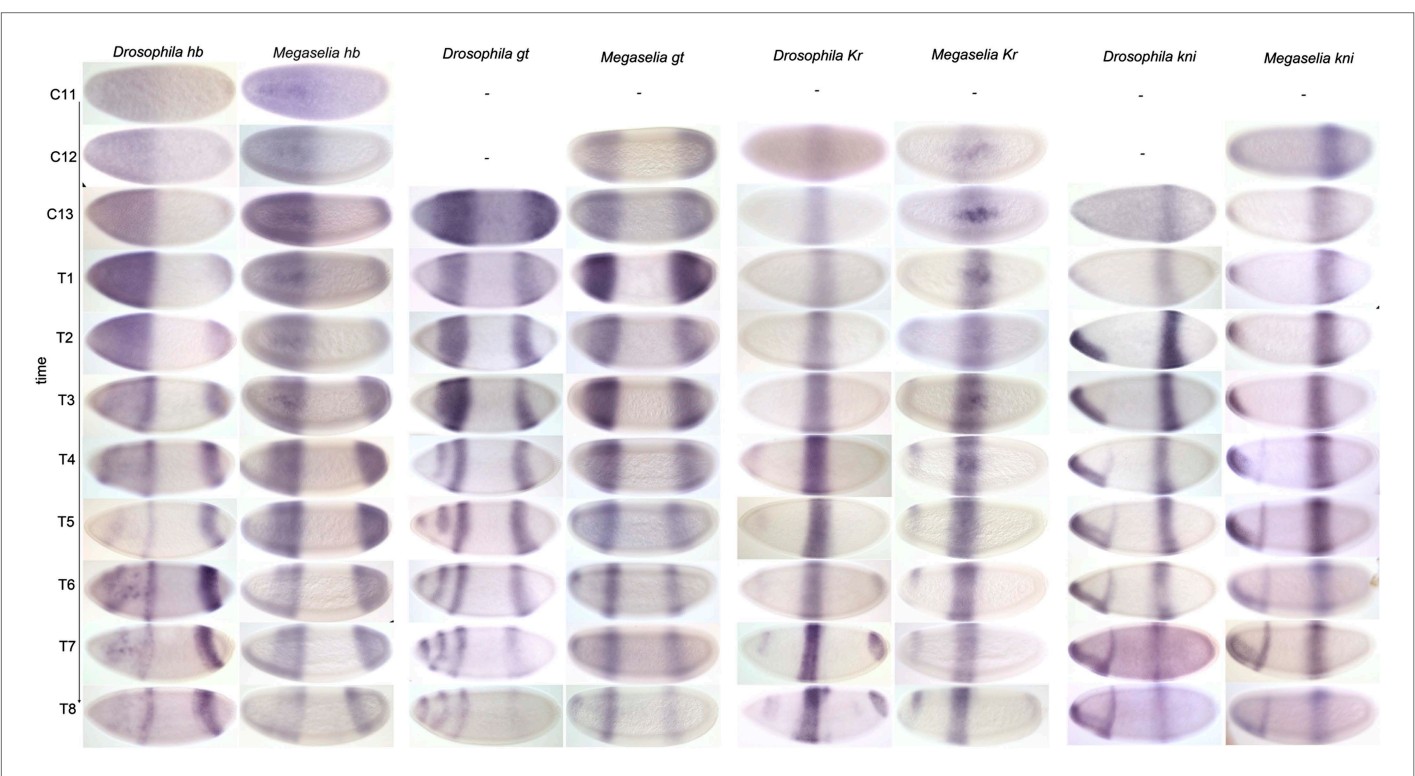

**Figure 11**. Expression data—trunk gap genes. Time series of mRNA expression patterns of trunk gap genes *hb, gt, Kr*, and *kni* are shown for *D. melanogaster* (left) and *M. abdita* (right). Expression patterns are visualised by colorimetric (enzymatic) in situ hybridisation using wide-field microscopy as described in *Crombach et al. (2012b)*. Early stages at the top, time progresses downwards. Embryo images show lateral views: anterior to the left, dorsal up. C11–13: cleavage cycles 11–13; T1–8: time classes subdividing C14A as defined in *Surkova et al. (2008b)* and *Wotton et al. (2014)*. *D. melanogaster* patterns for C14A were published in *Crombach et al. (2012a)* and are shown for comparison.

Around C12 or C13, a small posterior *hb* domain appears in *M. abdita*, extending to 90% A–P position as in *D. melanogaster* (*Figure 11*) (*Tautz et al., 1987*; *Crombach et al., 2012a*). By C14A-T2, the anterior expression border has shifted anteriorly to 84% A–P position, leading to an expansion of the posterior *hb* domain. In *D. melanogaster,* this domain retracts from the posterior pole around C14A-T2 (*Crombach et al., 2012a*). This retraction is delayed in *M. abdita.* Expression levels of *hb* start to drop in the posterior pole region around C14A-T6 but full retraction only occurs at C14A-T8 (*Figure 11*). At this stage, the posterior *hb* domain of *M. abdita* is positioned slightly further anterior (72–84 vs 75–85% A–P position), and is somewhat wider (12 vs 10%EL), than in *D. melanogaster* (*Crombach et al., 2012a*). This difference in domain width can be explained by the increased shift of the anterior boundary of posterior *hb* in *M. abdita* compared to *D. melanogaster* (18 vs 14% EL overall; see also *Figure 3*).

## giant (gt)

We first detect *gt* expression in *M. abdita* at C12 in an anterior stripe at 14–34% A–P position, and a posterior domain extending from 73% A–P position to the posterior pole (*Figure 11*). In *D. melanogaster*, *gt* expression is initiated at a similar time, during cycles C11 and C12 (*Jaeger et al., 2007*). During C13, the anterior domain is located more anteriorly in *M. abdita* than in *D. melanogaster* (15–35 vs 19–40% A–P position) but is of comparable width (20 vs 21%EL) (*Jaeger et al., 2007*; *Crombach et al., 2012a*). In *D. melanogaster,* this domain starts to split around C14A-T3 and forms two clearly resolved stripes at C14A-T4, and an additional expression domain (called the head patch) appears at the anterior pole at C14A-T5 (*Mohler et al., 1989*; *Kraut and Levine, 1991a*; *Crombach et al., 2012a*; *Becker et al., 2013*). These three anterior expression stripes persist until C14A-T8, and their posterior-most boundary shifts by about 3%EL to 37% A–P position at the onset of gastrulation (*Crombach et al., 2012a*; *Becker et al., 2013*). In *M. abdita*, *gt* expression starts to decrease in the anterior part of the anterior domain around C14A-T4 forming a single narrow stripe at C14A-T5 (*Figure 11*). At the same time, a domain comparable to the head patch of *D. melanogaster* appears at the anterior pole (*Figure 11*). In contrast to *D. melanogaster,* the posterior boundary of anterior *gt* expression does not shift significantly in *M. abdita* (see also *Figure 3*).

At C13, the anterior boundary of the posterior *gt* domain is at a slightly more anterior position in *M. abdita* than in *D. melanogaster* (72 vs 74% A–P position) (*Crombach et al., 2012a*). It also shifts further to the anterior by C14A-T8 in this species (10 vs 9% EL to 62 vs 65% A–P position). Furthermore, retraction from the posterior pole occurs slightly earlier in *M. abdita* than in *D. melanogaster*. Some *M. abdita* embryos (2 out of 13) show the first signs of retraction at C13, and by C14A-T1, the posterior boundary of posterior *gt* is found at 86% A–P position (*Figure 11*). In *D. melanogaster,* this domain only retracts at C14A-T1 with the posterior boundary located at 82% A–P position (*Crombach et al., 2012a*; *Becker et al., 2013*). Although the posterior *gt* domain of *M. abdita* remains wider for most of C14A, it narrows in both species to a width of about 8% EL by C14A-T8 (*Figure 11*). This is caused by a very rapid and strong shift of its posterior boundary during late C14A in *M. abdita,* which results in a larger overall anterior shift (18 vs 9% EL). During this process, the *M. abdita* boundary 'overtakes' its equivalent in *D. melanogaster* such that it is found further anterior (71 vs 73% A–P position) at C14A-T8 (*Figure 11*; see also *Figure 3*) (*Crombach et al., 2012a*; *Becker et al., 2013*).

## krüppel (Kr)

In *M. abdita,* we first detect *Kr* expression at C12, as is the case in *D. melanogaster* (*Knipple et al., 1985*; *Jaeger et al., 2007*). *Kr* appears in a central domain between 40 and 63% A–P position (*Figure 11*). This domain is much wider at this stage than in *D. melanogaster* (23 vs 12%EL), extending further to the anterior (by 8.6% EL) and to the posterior (by 2.6% EL) (*Knipple et al., 1985*; *Jaeger et al., 2007*). The posterior boundary of the central *Kr* domain shifts further in *M. abdita* than in *D. melanogaster* (12 vs 8% EL). In contrast, the anterior boundary shifts somewhat less (3% vs 6% EL). This results in a narrowing of the domain over time in both species (*Figure 11*). At all times, however, the central *Kr* domain remains wider (14 vs 10% EL at C14A-T8), and its final position at the onset of gastrulation lies further anterior (37–51 vs 43–53% A–P position), in *M. abdita* than in *D. melanogaster* (see also *Figure 3*) (*Crombach et al., 2012a*; *Becker et al., 2013*).

In addition to its central domain, *Kr* shows two additional domains during the blastoderm stage (*Knipple et al., 1985*; *Harding and Levine, 1988*). Neither of these are involved in gap gene cross-regulation. The head domain appears earlier in *M. abdita* than in *D. melanogaster* (C12 vs C14A-T4/T5),

while the posterior domain appears slightly later (C14A-T8 vs T6; *Figure 11*) (*Crombach et al., 2012a*; *Becker et al., 2013*). These domains exhibit similar expression dynamics in the two species, although both extend somewhat further into the central region of the embryo in *D. melanogaster* than in *M. abdita* (*Figure 11*).

## Knirps (kni)

The expression of *kni* in *M. abdita* is very similar to *D. melanogaster* (see *Figure 3*). We first detect *kni* expression at C12 in the anterior pole region extending to 7% A–P position and in an abdominal domain at 61–78% A–P position (*Figure 11*). While some studies have reported *kni* expression in *D. melanogaster* at C12 (*Rothe et al., 1989*, *1994*), more recent quantitative measurements could not detect *kni* before the mitotic division preceding the interphase of C13 (*Jaeger et al., 2007*). Throughout C13 and C14A, the abdominal *kni* domain shifts to the anterior and narrows progressively (*Figure 11*). The extent of the domain shift during C14A is slightly greater in *M. abdita* than in *D. melanogaster*: 4 vs 2% EL for the anterior and 8 vs 7% EL for the posterior boundary (*Crombach et al., 2012a*; *Becker et al., 2013*). The abdominal *kni* domain is also consistently wider and narrows more in *M. abdita* than in *D. melanogaster*.

In the anterior, expression at the pole remains at around 8–10% A–P position as in *D. melanogaster* (*Crombach et al., 2012a*). A head stripe domain appears around C14A-T3 or T4 at 24–27% A–P position (*Figure 11*). This is slightly further to the anterior than the equivalent head stripe in *D. melanogaster* (located at 27–30% A–P position), which appears at C14A-T4 (*Crombach et al., 2012a*). Expression in the head stripe is maintained up to C14A-T8.

## Appendix II: trunk gap gene RNAi cuticle phenotypes

In this section, we describe the gap gene RNAi cuticles of *M. abdita* and compare them to the published literature on gap gene mutants in *D. melanogaster*.

In *M. abdita*, *hb* knock-down (as previously published in *Stauber et al., 2000*) results in the deletion of the thorax, with additional defects in abdominal segments A1–8 and A8, and a reduced cephalopharyngeal skeleton, suggesting that the posterior gnathocephalon is also missing. This corresponds to a hypomorphic *hb* mutant phenotype in *D. melanogaster* (*Nüsslein-Volhard and Wieschaus, 1980*; *Jürgens et al., 1984*; *Lehmann and Nüsslein-Volhard, 1987*).

Other trunk gap gene knock-downs resemble *D. melanogaster* null mutants (*Figure 12*). *M. abdita gt* RNAi resulted in cuticles lacking abdominal segments A5–7 (*Figure 12B*) rather than just the denticle bands of abdominal segments A5–7 as in *D. melanogaster* (i.e., no naked cuticle is formed) (*Wieschaus et al., 1984a*; *Gergen and Wieschaus, 1986*; *Petschek et al., 1987*). Additionally, head segments are affected, and A8 is slightly reduced with filzkörper not fully present.

Following *kni* RNAi in *M. abdita* embryos, abdominal segments A1–7 merge (*Figure 12C*). In the most severe knock-down phenotypes,

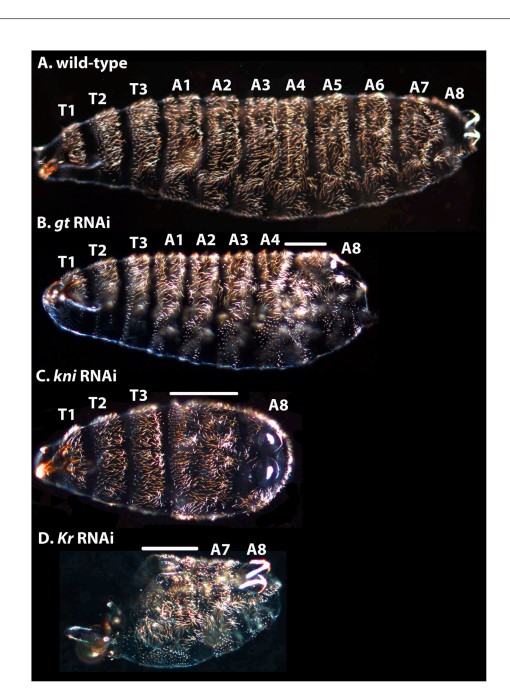

**Figure 12**. Cuticle phenotypes resulting from RNAi knock-down of gt, kni, and Kr in *M. abdita*. (**A**) A wild-type cuticle is shown for comparison to severe phenotypes in *gt* (**B**), *kni* (**C**), *and Kr* (**D**) RNAi-treated embryos. We tentatively assign segment identity to persisting abdominal segments based on their relative spatial order, the position of expression domains of the knocked-down genes in the blastoderm (see Appendix I), and comparison to the corresponding cuticle phenotypes in *D. melanogaster*. T1–T3; thoracic segments, A1–A8; abdominal segments; solid line indicates fused segments. Cuticles are shown in lateral (**A**, **B**), ventral (**C**), or ventro-lateral (**D**) view. Anterior is to the left. *hb* RNAi knock-down phenotypes were previously published in *Stauber et al. (2000)*.

the fused segments are marked by a single, irregular denticle field. Intermediate phenotypes may exhibit a few remaining discernible abdominal denticle belts. In some *kni* RNAi cuticles, more posterior markers as well as the posterior portion of the T3 denticle belt were distorted as well. Head segments appear unaffected. This severe *kni* RNAi phenotype is very similar to the null mutant *kni* phenotype in *D. melanogaster* (**Nüsslein-Volhard and Wieschaus, 1980**; **Jürgens et al., 1984**; **Nauber et al., 1988**).

In severely affected *M. abdita Kr* knock-down cuticles, only the telson and A8, plus one or two additional abdominal segments are formed (putative A7, and sometimes A6; *Figure 12D*). Other abdominal segments and all thoracic segments are either missing or fused, and head involution fails to occur. Head segments were strongly affected. Unlike in *D. melanogaster*, where A6 is duplicated as a mirror image in *Kr* mutants (**Nüsslein-Volhard and Wieschaus, 1980**; **Wieschaus et al., 1984b**), we cannot identify clear instances of local inversions or duplications in *M. abdita.* However, it is possible that we have missed these phenotypes, due to the less obvious polarity of the segmental denticle patterns in *M. abdita* compared to *D. melanogaster.*

## Acknowledgements

We would like to thank Astrid Hoermann for her contribution to virgin collection and embryo fixation for the *Drosophila cad* germ-line clones. We thank Benjamin Krinsky (University of Chicago) for conducting initial qualitative RNA in situ hybridizations with *Megaselia* gap gene probes. Brenda Gavilán (Universitat Autònoma de Barcelona) and Núria Bosch Guiteras (Universitat Pompeu Fabra) helped with the maintenance of the *Megaselia* culture in the Jaeger lab.

## Additional information

### Funding

| Funder | Grant reference number | Author |
| --- | --- | --- |
| Ministerio de Economía y Competitividad | MEC/EMBL Agreement/ BFU2009-10184/ BFU2012-33775/ SEV-2012-0208 | Karl R Wotton, Eva Jiménez-Guri, Anton Crombach, Hilde Janssens, Anna Alcaine-Colet, Johannes Jaeger |
| Agència de Gestió d'Ajuts Universitaris i de Recerca | SGR Grant 406 | Johannes Jaeger |
| European Commission | FP7-KBBE-2011-5/289434 | Anton Crombach, Johannes Jaeger |
| National Science Foundation | IOS-0719445/IOS-1121211 | Steffen Lemke, Urs Schmidt-Ott |

The funders had no role in study design, data collection and interpretation, or the decision to submit the work for publication.

### Author contributions

KRW, Conception and design, Acquisition of data, Analysis and interpretation of data, Drafting or revising the article; EJ-G, Acquisition of data, Analysis and interpretation of data, Drafting or revising the article; AC, Analysis and interpretation of data, Drafting or revising the article; HJ, Acquisition of data, Analysis and interpretation of data; AA-C, Analysis and interpretation of data; SL, US-O, Analysis and interpretation of data, Contributed unpublished essential data or reagents; JJ, Conception and design, Analysis and interpretation of data, Drafting or revising the article

### Author ORCIDs

Karl R Wotton, http://orcid.org/0000-0002-8672-9948
Eva Jiménez-Guri, http://orcid.org/0000-0002-9592-1077
Johannes Jaeger, http://orcid.org/0000-0002-2568-2103

## Additional files

### Supplementary files

• Supplementary file 1. Maternal co-ordinate and terminal gap gene mRNA expression dataset for *M. abdita*. This file contains a table with numbers of embryos in our dataset for maternal co-ordinate and terminal gap gene mRNA expression in *M. abdita*.

• Supplementary file 2. Boundary positions/shifts and domain widths/overlaps contain tables with numerical comparisons of expression data between *M. abdita* and *D. melanogaster*. (**A**) Comparison of expression data between *M. abdita* and *D. melanogaster*: gap domain boundary positions. (**B**) Comparison of expression data between *M. abdita* and *D. melanogaster*: gap domain widths. (**C**) Comparison of expression data between *M. abdita* and *D. melanogaster*: gap domain boundary shifts. (**D**) Comparison of expression data between *M. abdita* and *D. melanogaster*: gap domain overlaps.

• Supplementary file 3. *Megaselia* segmentation gene expression.

• Supplementary file 4. RNAi dataset.

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
