## [Decision Letter]

Thank you for sending your work entitled “Quantitative System Drift Compensates for Altered Maternal Inputs to the Gap Gene Network of the Scuttle Fly *Megaselia abdita*” for consideration at *eLife*. Your article has been favorably evaluated by Diethard Tautz (Senior editor), a Reviewing editor, and 2 reviewers.

There is a uniform agreement that the study is important and should be published, as it provides the first comprehensive quantitative analysis of gap gene expression in non-drosophilid dipteran using a functional approach (RNAi).

The authors claim that the gap gene domains of *Megaselia* are initially different from *Drosophila*, but in the course of late blastoderm start to converge to the *Drosophila* pattern. Although this does not apply for anterior *giant* and anterior *hunchback*, convergence can be seen for some posterior domains. As a nice addition to the functional work with *Megaselia,* the authors present a germ line clone analysis for *caudal* in *Drosophila* which shows that the absence of maternal *caudal* in *Drosophila* leads to shifts in gap gene domains, which are similar to those observed in *Megaselia*.

The authors conclude that the gap gene network of *Megaselia* has the same structure as that of *Drosophila* and that the observed differences are due to quantitative changes in the strength of gap gene interactions. Thus, genetic drift yields quantitatively different (parametric) differences between nodes in the gap gene network, but the overall network connectivity and sign of the regulatory interaction (activating or inhibiting) remains largely intact.

Revision required for publication:

1) The manuscript is not easy to read, in particular the Results and discussion part. Here, the text tends to be redundant and should be shortened. Box 1 is meant to help the reader, but is not easy to understand mainly because the sentences are difficult to read. The authors may consider whether presenting a numbering scheme for boundaries in [Supplementary-material SD2-data] would be a better formalism to describe the spatial patterning features in the text.

2) The caudal data should be discussed and clarified. In particular, it would be good to check whether hetero- or homozygosity of zygotic *caudal* alone affects *gt* and *kni* expression (this might be in the literature). If this is not the case, then it is likely that the observed shifts are due to the absence of maternal *caudal*.

---

## [Author Response]

*1) The manuscript is not easy to read, in particular the Results and discussion part. Here, the text tends to be redundant and should be shortened*.

We have done our best to clarify and shorten the Results and discussion section by reducing redundancy wherever possible. Implemented changes include:

Removing redundant descriptions of methodology.

Removing redundant descriptions of regulatory mechanisms.

Consolidating summary paragraphs at the end of each subsection.

We believe that these efforts improve the readability of our manuscript considerably, while preserving the hierarchical overall structure of the Results and discussion section. This structure is designed to provide the reader with a large amount of detailed information, while still retaining a focus on the big picture.

Box 1
*is meant to help the reader, but is not easy to understand mainly because the sentences are difficult to read*.

We have edited Box 1 to make it more readable and concise, as requested.

*The authors may consider whether presenting a numbering scheme for boundaries in*
[Supplementary-material SD2-data]
*would be a better formalism to describe the spatial patterning features in the text*.

We are reluctant to use a boundary-numbering scheme as in [Supplementary-material SD3-data] to identify expression features. While this would indeed make the text more concise, our use of explicit descriptions (e.g. “posterior boundary of the anterior *hb* domain”) has the advantage of being unambiguous and directly interpretable without use of a look-up table to remember which numbered boundary is which. In addition, it is consistent with boundary descriptions in previously published studies.

*2) The caudal data should be discussed and clarified. In particular, it would be good to check whether hetero- or homozygosity of zygotic caudal alone affects gt and kni expression (this might be in the literature). If this is not the case, then it is likely that the observed shifts are due to the absence of maternal caudal*.

We have clarified our description of the design and interpretation of the *cad* germ line clone experiment. Without quantitative measurements of absolute mRNA levels, we cannot distinguish between the effects of lacking maternal expression, or altered levels of zygotic *cad* in *M. abdita* compared to *D. melanogaster.*

This is now stated explicitly in the Results and Discussion, and Materials and Methods sections. What our germ line experiments in *Drosophila* establish is that altered expression of *cad* can explain delayed shifts of posterior gap domains. Whether this is governed by maternal or zygotic contributions to *cad* is not directly relevant at this level of argumentation. A proper quantitative investigation of these contributions goes beyond the scope of the current manuscript, which focuses on the downstream interactions that cause subsequent changes in shift dynamics.

In addition to the changes in response to editorial and reviewers’ feedback described above, we have also revised our supplementary information according to the discussion we had with the *eLife* editorial office. Detailed descriptions of gap gene expression patterns and cuticle phenotypes in *Megaselia* are now provided as two appendix sections (with additional Figures 10, 11 and 12) in the main paper, as requested.